# Modeling the influence of carbon branching structure on SOA formation via multiphase reactions of alkanes

Azad Madhu[1], Myoseon Jang[1], Yujin Jo[1]

[1]Engineering School of Sustainable Infrastructure and Environment, University of Florida, Gainesville, 32608, United States of America

*Correspondence to*: Myoseon Jang (mjang@ufl.edu)

**Abstract.**

Branched alkanes represent a significant proportion of hydrocarbons emitted in urban environments. To accurately predict the SOA budgets in urban environments, these branched alkanes should be considered as SOA precursors. However, the potential to form SOA from diverse branched alkanes under varying environmental conditions is currently not well understood. In this study, the Unified Partitioning Aerosol Phase Reaction (UNIPAR) model is extended to predict SOA formation via the multiphase reactions of various branched alkanes. Simulations with the UNIPAR model, which processes multiphase partitioning and aerosol phase reactions to form SOA, require a product distribution predicted from an explicit gas kinetic mechanism, whose oxygenated products are applied to create volatility-reactivity based $\alpha_i$ species array. Due to a lack of practically applicable explicit gas mechanisms, the prediction of the product distributions of various branched alkanes was approached with an innovative method that considers carbon lengths and branching structures. The $\alpha_i$ array of each branched alkane was primarily constructed using an existing $\alpha_i$ array of the linear alkane with the nearest vapor pressure. Generally, the vapor pressures of branched alkanes and their oxidation products are lower than those of linear alkanes with the same carbon number. In addition, increasing the number of alkyl branches can also decrease the ability of alkanes to undergo autoxidation reactions that tend to form low-volatility products and significantly contribute to alkane SOA formation. To account for this, an autoxidation reduction factor, as a function of the degree and position of branching, was applied to the lumped groups which contain autoxidation products. The resulting product distributions were then applied to the UNIPAR model for predicting branched alkane SOA formation. The simulated SOA mass was compared to SOA data generated under varying experimental conditions (i.e., $NO_x$ levels, seed conditions, and humidity) in an outdoor photochemical smog chamber. Branched alkane SOA yields were significantly impacted by $NO_x$ levels but insignificantly impacted by seed conditions or humidity. The SOA formation from branched and linear alkanes in diesel fuel was simulated to understand the relative importance of branched and linear alkanes in a wide range of carbon numbers. Overall, branched alkanes accounted for a higher proportion of SOA mass than linear alkanes due to their higher contribution to diesel fuel.

 **1 Introduction**

Secondary Organic Aerosol (SOA) in the atmosphere, formed via the atmospheric oxidation of various precursor hydrocarbons (HCs), serves a considerable role in climate, cloud formation, and human health (Nel, 2005; Shrivastava et al., 2017; World Health, 2016; Epa, 2019). Precursor HCs found in ambient air are emitted through a variety of both anthropogenic and biogenic sources. Alkanes are one of the major classes of precursor HCs typically found in urban environments, emitted from anthropogenic sources such fossil fuels, personal care products, paints, and pesticides (Li et al., 2022; Wu et al., 2019). In addition to anthropogenic sources, plant wax has also been identified to be a significant source of alkanes present in SOA (Alves et al., 2012). Alkanes do not represent a large proportion of biogenic emissions relative to other biogenic VOCs but they tend to be large, low-volatility compounds which can still significantly contribute to SOA (Männistö et al., 2023; Lyu et al., 2017). Several recent gas sampling studies have identified alkanes, alongside aromatics, as one of the two dominant sets of HCs measured in urban environments (Massolo et al., 2010; Song et al., 2019; Xuan et al., 2021; Zhao et al., 2020). For example, Song et al. (2019) reports from their review of various VOC sampling studies that alkanes represent between 40.3% to 74.4% of VOCs in a set collected from Houston, Mexico, and various urban cities in China. Within the set of alkanes in their study, Song et al. (2019) finds that branched alkanes represented 33% of alkanes sampled in Langfang, China. This is consistent with Caplain et al.'s (2006) study which reported that linear and substituted alkanes represent significant proportions of both gasoline and diesel fuel exhaust at 6-18% and 18-31%, respectively. Additionally, recent regional modeling studies have shown that alkanes are also one of the dominant sources of secondary organic aerosol (SOA) formation in urban environments (Li et al., 2022; Jo et al., 2023; Yang et al., 2019). Within the set of alkanes, branched alkanes represent larger proportions of both gasoline and diesel fuels compared to linear alkanes (Gentner et al., 2012). Furthermore, intermediate-volatility organic compounds (IVOCs) in the atmosphere are largely composed of an unresolved complex mixture, which is likely to consist of a variety of branched alkane isomers (Tkacik et al., 2012).

Currently available mechanisms of branched alkanes are either overly simplistic, with a limited set of oxidation products (Lim and Ziemann, 2009), or overly complex, with too many products to be incorporated into an SOA model (i.e. GECKO-A). Some studies have managed to model the SOA formation of various branched alkanes using the GECKO-A model (Aumont et al., 2013; La et al., 2016). However, because the number of oxidation products in the GECKO-A model increases exponentially as the number of carbons of the HC precursor increases, these studies had significant limitations as they tried to reduce the number of products to a manageable value. For example, La et al. (2016) reduced the number of products considered in the model by limiting the experimental time to a maximum of 1 hour. Aumont et al. (2013) reduced the number of precursors considered by only considering the High $NO_x$ condition, a limitation which they acknowledge is a "severe simplification." For application to a regional model, parameters generated from chamber models must be applicable for a variety of $NO_x$ conditions for the whole range of oxidation products generated throughout a long oxidation time.

In recent chamber study by Madhu et al. (2023), the SOA formation of a series of linear alkanes was simulated for a wide range of carbon numbers using the UNIfied Partitioning Aerosol Reaction (UNIPAR) model, which simulate SOA mass via

multiphase reactions of HCs. The UNIPAR model utilizes $\alpha_i$ array generated from explicitly predicted gas oxidation products. The volatility-reactivity based lumped species in the array are applied to multiphase partitioning and aerosol phase reactions

to form SOA mass (Choi and Jang, 2022; Han and Jang, 2022; Im et al., 2014; Yu et al., 2021; Zhou et al., 2019). In order to predict SOA mass originating from linear alkanes at different carbon lengths, an incremental volatility coefficient (IVC) was used to predict the $\alpha_i$ arrays of larger linear alkanes without explicit mechanisms (Madhu et al., 2023). In their model simulation, the importance of low-volatility autoxidation products in linear alkane SOA formation was demonstrated.

The molecular structure of an alkane serves an important role in its capability to form SOA mass. Linear and branched alkanes

tend to have similar oxidation products, with the distinction that branched alkane products tend to be more volatile at the same precursor carbon number (Lim and Ziemann, 2009). Additionally, the presence of some alkyl branches can decrease the ability of alkanes to participate in autoxidation reactions and increase the decomposition of intermediate products, which further reduces SOA yields by decreasing the amount of low-volatility oxidation products formed. As $NO_x$ decreases in polluted urban areas, the contribution of lowly volatile autoxidation products becomes relatively more important in SOA formation (Pye

Havala et al., 2019; Praske et al., 2018). In this study, the UNIPAR model is used to simulate the SOA formation from the multiphase reactions of branched alkanes. Due to a lack of practically applicable explicit gas mechanisms, the product distributions of branched alkanes were constructed primarily using the product distributions of linear alkanes with the nearest vapor pressure. To account for the reduction of probability for autoxidation reactions, an autoxidation reduction factor, which is a function of the degree of branching and position, was applied to the construction of lumped groups that include autoxidation

products. Predicted $\alpha_i$ arrays for various branched alkanes (Isododecane, 2,6,10-Trimethyldodecane, 2,2,4,4,6,8,8-Heptamethylnonane, and 2,4,6,10-Tetramethylpentadecane) were used to simulate SOA formation, which was compared to chamber generated SOA data. The sensitivity of branched alkane SOA yields to humidity, NOx conditions, seed conditions, and branching structure at given simulation conditions was projected. Additionally, a simulation of SOA formation from branched and linear alkanes found in diesel was used to determine their relative significance as SOA precursors.

**2 Experimental section**

Alkane SOA was produced from photooxidation of a set of branched alkanes [Isododecane (Sigma-Aldrich; 80%), Trimethyldodecane (Sigma-Aldrich; 99%), Heptamethylnonane (Sigma-Aldrich; 98%), Tetramethylpentadecane (Fisher Scientific; 95%)] using the UF-APHOR dual chamber (52 $m^3$ each) located at the University of Florida. The detailed description of the operation of the large outdoor smog chamber can be found in a previous study (Im et al., 2014). Table 1

summarizes the experimental conditions of outdoor chamber experiments. Precursor alkane hydrocarbons (HCs) were injected into chamber through evaporation via heating. $CCl_4$ (Sigma-Aldrich; ≥99.5%), was introduced into the chamber as a non-reactive gas which is used to measure chamber dilution. HONO was used as a source of hydroxyl radicals in the chamber. HCs, HONO, and NO (2% in $N_2$, Airgas Inc., USA) and inorganic seed were introduced into the smog chamber before sunrise. Experiments were performed under two different NOx levels (high $NO_x$: HC/$NO_x$ <5 ppbC/ppb; low $NO_x$: HC/$NO_x$ >10

ppbC/ppb) and three different seed conditions (without seed, sulfuric acid, and ammonium sulfate). Concentrations of gas phase HCs and $CCL_4$ were measured using a GC-FID (7820A, Agilent Technologies, Inc., USA). Concentrations of ozone and $NO_x$ within the chamber were measured using a photometric ozone analyzer (400E, Teledyne Technologies, Inc., USA) and a chemiluminescence $NO/NO_x$ analyzer (T201, Teledyne Technologies, Inc., USA), respectively. Experiments using inorganic seed employed a Particle into liquid sampler (ADISO 2081, Applikon Inc., USA) integrated with Ion Chromatography

(Compact IC 761, Metrohm Inc., Switzerland) (PILS-IC) to measure inorganic ion concentrations within the chamber. The size distribution of particles within the chamber was measured using a scanning mobility particle sizer (SMPS 3080, TSI Inc., USA).

    Previous studies that have measured the density of alkane SOA have found a range from 1 to 1.4 $g/cm^3$ (Li et al., 2020; Li et al., 2022; Lim and Ziemann, 2009; Loza et al., 2014). Aerosols from each alkane experiment in this study were assumed to

have a density of 1.2 $g/cm^3$. A hygrometer (CR1000 measurement and control system, Campbell Scientific Inc., USA) was used to measure meteorological factors (temperature, relative humidity (RH) and an ultraviolet radiometer (TUVR, Eppley Laboratory Inc., USA) was used to measure sunlight intensity. An organic carbon/elemental carbon analyzer (OC/EC model 4, Sunset Laboratory Inc., USA) was used to measure the concentration of organic carbon in aerosol every 50 minutes. The OC/EC used a non-dispersive infrared detector (NDIR) which measured OC using thermal optical transmittance. The

concentration of organic matter in aerosol (OM, $\mu g\ m^{-3}$) was then calculated based on the OC concentration predicted by the UNIPAR model and an OM to OC ratio. The OM to OC ratio of SOA from alkane species decreased as the chain length increased. The concentrations of OM measured from the chamber were corrected for chamber dilution using a dilution factor and for particle wall loss to the chamber wall using a particle loss factor. An aerosol chemical speciation monitor (ACSM, Aerodyne Research Inc., USA) was used to measure the aerosol composition (sulfate, nitrate, ammonium, and OM). The

compositions obtained from the ACSM were compared with measurements from the OC/EC and the PILS-IC. SOA yields ($Y$) were then calculated as the final measured concentration of OM divided by the total consumption of HC precursors. In order to characterize the chemical functional distribution of SOA, chamber-generated SOA was collected on a silicon disk ($13 \times 2$ mm, Sigma-Aldrich, USA) using a home-built impactor and analyzed using the FTIR spectrometer (Nicolet iS50, Thermo Fisher Inc., USA) in transmission mode. The FTIR disc was weighed using an analytical balance before and after particle

impaction to measure particle mass collected.

Table 1. Summary of experimental conditions and observed data for experiments performed in the UF-APHOR outdoor chamber.

| Label | Date | HC name | Initial NOx (PPB) | Initial HONO (PPB) | HC[a] initial (PPB) | Seed[b] | HC Consumed[c] (PPB) | Final OC ($\mu g/m^3$) | SOA yield[d] | Comments |
|---|---|---|---|---|---|---|---|---|---|---|
| C12A | 07/08/22 | Isododecane | 911 | 100 | 220 | None | 76.5 | 0.7 | 0.001 | Fig.2 |
| C12B | 06/07/22 | N-dodecane | 829 | 97 | 159 | None | 173.4 | 19.0 | 0.024 | Madhu et al. (2023) Fig. 3 |
| C12C | 06/07/22 | N-dodecane | 331 | 200 | 135 | None | 131.3 | 123.2 | 0.206 | Madhu et al. (2023) FTIR, Fig. 4 |
| C15A | 07/18/22 | Trimethyldodecane | 936 | 140 | 220 | None | 162.0 | 103.8 | 0.110 | Fig.2 |
| C15B | 08/04/22 | Trimethyldodecane | 607 | 107 | 262 | SA | 161.6 | 141.2 | 0.150 | Fig.2 |
| C15C | 02/17/22 | N-pentadecane | 665 | 117 | 202 | None | 125.3 | 117 | 0.285 | Madhu et al. (2023) Fig. 3 |
| C16A | 09/02/22 | Heptamethylnonane | 433 | 80 | 170 | None | 60.0 | 3.5 | 0.006 | Fig.2 |
| C16B | 09/02/22 | Heptamethylnonane | 205 | 60 | 165 | None | 63.0 | 1.7 | 0.003 | Fig.2 |
| C16C | 04/14/23 | Heptamethylnonane | 120 | 90 | 100 | AS | 55.3 | 26.1 | 0.051 | Fig.2 |
| C16D | 04/14/23 | Heptamethylnonane | 120 | 90 | 100 | SA | 60.6 | 25.7 | 0.046 | Fig.2 |
| C19A | 07/29/22 | Tetramethylpentadecane | 260 | 110 | 195 | None | 147.7 | 447.5 | 0.276 | Fig.2 FTIR, Fig. 4 |
| C19B | 08/09/22 | Tetramethylpentadecane | 251 | 100 | 163 | SA | 141.4 | 267.5 | 0.172 | Fig.2 FTIR, Fig. 4 |
| C19C | 08/09/22 | Tetramethylpentadecane | 232 | 100 | 163 | NS | 140.1 | 389.7 | 0.253 | Fig.2 |
| C19D | 03/30/23 | Tetramethylpentadecane | 229 | 39 | 39 | NS | 30.1 | 79.0 | 0.239 | Fig.2 |
| C19E | 03/30/23 | Tetramethylpentadecane | 143 | 73 | 39 | NS | 36.1 | 149.6 | 0.377 | Fig.2 |

[a] Initial concentrations of Tetramethylpentadecane were calculated using a 95% injection efficiency as this compound is too low volatility to be measured by the GC-FID. [b] Experiments were performed with no seed (none) and sulfuric acid seed (SA). [c] Values for the amount of Tetramethylpentadecane consumed in each experiment are reported based on model simulations seen in Fig. S3 as this compound is too low volatility to be measured by the GC-FID. [d] Yield was calculated as the ratio between the concentration of the final measured SOA mass ($\mu g/m^3$) and the concentration of precursor alkane consumed ($\mu g/m^3$).

## 3 Model description

The efficacy of the UNIPAR model has been demonstrated for simulating SOA formation in photochemical chamber experiments from a variety of HC classes, including aromatic HCs (Im et al., 2014; Zhou et al., 2019; Han and Jang, 2022), monoterpenes (Yu et al., 2021b), and isoprene (Beardsley and Jang, 2016). Furthermore, UNIPAR has been recently integrated into the CAMx regional scale model and used to demonstrate SOA formation from various HCs including isoprene, terpenes, aromatics, and linear alkanes (Jo et al., 2023; Yu et al., 2022). In this study, UNIPAR model parameters were developed for the simulation of SOA formation from branched alkanes. The UNIPAR model simulates SOA mass formation via the multiphase reactions of HC precursors including gas ($g$), organic ($org$) and inorganic ($inorg$) phases. Typically, the UNIPAR model employs a near-explicit gas oxidation mechanism which is used to create a product distribution for each HC precursor by lumping products into an $\alpha_i$ array consisting of 48 groups according to their volatility and reactivity. An approach using product structure-based $\alpha_i$ array renders the ability to simulate SOA formation via multiphase partitioning and in-particle chemistry that forms oligomeric products. In addition to SOA mass, SOA's chemical and physical characteristics, such as oxygen-to-carbon (O:C) and aerosol viscosity were simulated in the model. Detailed descriptions of lumping criteria and the mass-based stoichiometric coefficient ($\alpha_i$) of lumped group $i$ as a function of $NO_x$ levels and the degree of aging can be found in previous studies (Madhu et al., 2023; Han and Jang, 2022; Choi and Jang, 2022; Yu et al., 2021b; Zhou et al., 2019). Because of a lack of practically applicable gas oxidation mechanisms for the variety of branched alkanes found in the atmosphere, a novel method was used to create $\alpha_i$ arrays for each branched alkane. Individual branched alkane $\alpha_i$ arrays were constructed primarily using existing $\alpha_i$ arrays, developed by Madhu et al. (2023), of the linear alkanes with the nearest vapor pressures. To account for the reduction of autoxidation products created by branched alkanes, an autoxidation reduction factor, which is a function of the degree and position of branching, was applied to lumped groups which contain autoxidation products. Figure 1 shows the UNIPAR model frame. In the presence of inorganic seed, the SOA mass is simulated via three paths: OM produced via multiphase partitioning of organic products ($OM_P$), aerosol phase reactions of organic species to form $OM_{AR}$ via oligomerization in the $org$ phase, and reactions in the wet $inorg$ phase which also form $OM_{AR}$ (acid-catalyzed oligomerization and organosulfate (OS) formation).

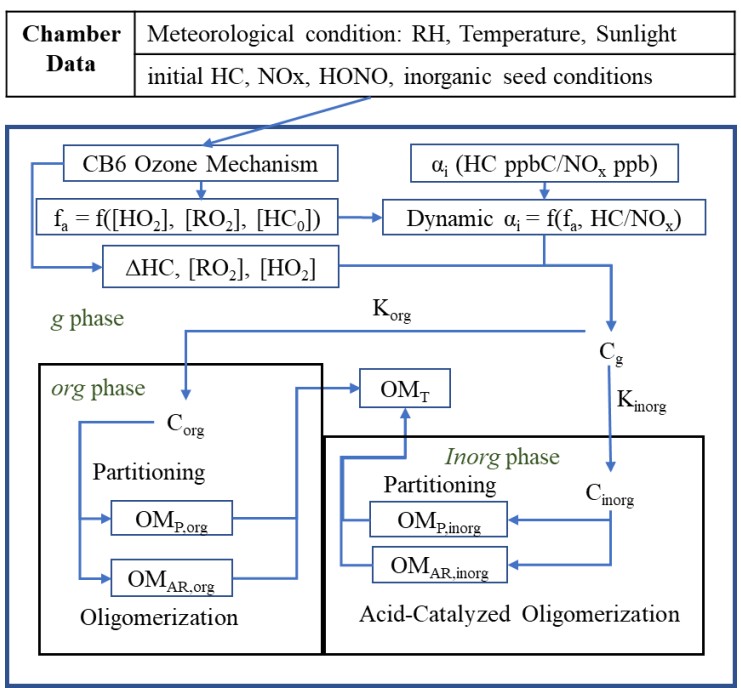

Figure 1. A simplified scheme of the UNIPAR model used in this study. $[HC]_0$ represents the initial hydrocarbon (HC) concentration. Chamber generated data is used to set initial conditions and the meteorological condition for the gas simulation. The CB6 Ozone mechanism is used to simulate the consumption of HC ($\Delta HC$) and the concentrations of hydroperoxide radical ($[HO_2]$) and organic peroxyl radical ($[RO_2]$). The dynamic mass-based stoichiometric coefficients (dynamic $\alpha_i$) of lumped species $i$ are calculated as a function of $HC/NO_x$ and the aging factor ($f_a$). $f_a$ is represented as a function of $[HO_2]$, $[RO_2]$, and $[HC]_0$ (Zhou et al., 2019). The gas, organic, and inorganic phases are represented by the subscripts $g$, $org$, and $inorg$, respectively. $K_{org}$ and $K_{inorg}$ represent the partitioning coefficients of lumped species to the $org$ phase and $inorg$ phase, respectively. $C_{org}$ and $C_{inorg}$ represent the concentrations of lumped species in the $org$ and $inorg$ phases, respectively. $OM_{P,org}$ and $OM_{P,inorg}$ represent the mass of organic matter (OM) present in $org$ and $inorg$ phases, respectively, due to partitioning. $OM_{Ar,org}$ and $OM_{AR,inorg}$ represent the OM formed in the $org$ phase due to in-particle chemistry such as oligomerization, and $inorg$ phase due to acid-catalyzed oligomerization and organosulfate formation (Beardsley and Jang, 2016; Im et al., 2014; Zhou et al., 2019). $OM_T$ represents the total SOA mass formed.

### 3.1 UNIPAR model inputs

The parameter inputs to the UNIPAR model for each precursor HC include the equations for $\alpha_i$ and lumped species' physicochemical parameters, such as molecular weight (MW), O:C ratios, and hydrogen bonding (HB). The model inputs associated with each individual experiment include the consumption of HC ($\Delta HC$); the concentrations of alkyl peroxide radical ($RO_2$) and hydroperoxyl radical ($HO_2$); concentrations of ionic species (sulfate and ammonium ions); temperature, and humidity. The measured sunlight profile for each experiment is linked to the gas oxidation of each HC yielding $\Delta HC$, $[RO_2]$,

and [HO₂]. Gas simulation was performed in the box model platform of Dynamically Simple Model of Atmospheric Chemical Complexity (DSMACC(Emmerson and Evans, 2009)) interfaced with the kinetic pre-processor (KPP). The predetermined mathematical equations for stoichiometric coefficients were constructed for linear alkanes by explicit gas mechanism as described in the following section 3.2. For the simulation of SOA mass, HC consumption of branched alkane was simulated with Carbon Bond 6 mechanism (CB6r3(Emery et al., 2015)). Further details about the CB6 mechanism used in this study can be found in section S1.

## 3.2 Gas mechanisms

The atmospheric oxidation of alkanes begins with the reaction with an OH radical, followed by the addition of $O_2$ to form peroxyl radicals. In the presence of NOx, these peroxyl radicals can form alkoxy radicals or organonitrate (Finlayson-Pitts and Pitts, 2000). In addition, peroxyl radicals can create hydroperoxides via the reaction with $HO_2$ radicals. In the previous literature (Crounse et al., 2013; Bianchi et al., 2019; Roldin et al., 2019), the terpene peroxyl radicals are capable of undergoing autoxidation reactions to form low volatility products. These reactions have been previously modeled (Pye Havala et al., 2019; Xavier et al., 2019). In the recent modelling study, Madhu et al. (2023), applies autoxidation mechanisms to improve the prediction of a series of linear alkane SOA. In their study, the product distribution of linear alkanes relatively in small carbon lengths (C9-C12) were predicted using explicit gas mechanisms which consisted of their respective Master Chemical Mechanism (MCM) and newly added autoxidation mechanisms. The product distributions, associated with stoichiometric coefficients, of $\alpha_i$ array were extrapolated to larger linear alkanes which do not currently have MCM mechanisms using an Incremental Volatility Coefficient (IVC). The feasibility of the $\alpha_i$ arrays generated with the IVC approach was demonstrated to predict chamber generated SOA data. For branched alkanes, explicit gas mechanisms are not currently available for the practical application to the UNIPAR SOA model, as described in the Introduction section. In this study, to produce product distributions for the variety of branched alkanes, the pre-existing linear alkane $\alpha_i$ arrays are extrapolated to branched alkanes as seen in section 3.4 below.

## 3.3 Lumping structure of UNIPAR model

As discussed in section 3.2, the $\alpha_i$ array of branched alkanes in the UNIPAR model is inherited from linear alkanes.
The lumping structure of the UNIPAR model, along with a dynamic $\alpha_i$ array which considers aging, has been developed in previous studies (Zhou et al., 2019; Han and Jang, 2020; Yu et al., 2021b). The $\alpha_i$ array is constructed based on products' volatility-reactivity characteristics. The $\alpha_i$ array consists of 6 different reactivity levels (very fast (VF), fast (F), medium (M), slow (S), partitioning only (P), and multi-alcohol (MA)) and 8 different volatility levels (1E-08, 1E-06, 1E-05, 1E-04, 1E-03, 1E-02, 1E-01, and 1.0 mmHg) based on vapor pressure which represent 48 species. The 8 volatility levels have enthalpy of vaporization values 140E+3, 106E+3, 96E+3, 89E+3, 82E+3, 58E+3, 58E+3, and 58E+3 J/mol, respectively. Further description regarding the basis for the enthalpy of vaporization values can be found in section S2. During the process, each

non-radical gas oxidation product of a specific precursor is lumped into one of the 48 species. During SOA simulations, all oxidation products within a specific lumped group with undergo partitioning, or particle phase reactions, as single species. For example, consider the compound 3-dodecanol, a product of n-dodecane photooxidation. 3-dodecanol has a calculated vapor pressure of 5.1E-3 mmHg (Myrdal and Yalkowsky, 1997; Zhao et al., 1999; Stein and Brown, 1994). Because the vapor pressure is smaller than that of group 7 (1E-2 mmHg) and larger than that of group 6 (1E-3 mmHg), 3-dodecanol is lumped into volatility group 6. Due to a lack of functional groups which promote particle phase reactions (i.e. ketone and aldehyde), 3-dodecanol is lumped into reactivity group P and will only participate in partitioning. This process is performed to lump each non-radical gas oxidation product, with a sufficiently low vapor pressure, into one of the 48 lumped groups. If group 6P contains only 3-dodecanol, then the α value of that group represents the units (i.e. μg/m$^3$) of 3-dodecanol produced per unit of precursor HC consumed. If group 6P has multiple compounds, then then the α value of that group represents that the sum of the units produced of all products classified under 6P per unit of precursor consumed. After the initial lumping process, the amounts of each gas oxidation species produced per unit of HC consumed is extracted via a standardized gas simulation. This gas simulation predicts gas oxidation product concentrations for a given initial precursor concentration at various NOx levels under a sunlight file near the middle of summer (06/23/18). These simulation results produce lumped arrays for 4 different conditions: high NOx fresh, high NOx aged, low NOx fresh, and high NOx aged. The α value of each group is represented as a polynomial equation which is a function of the HC ppbC/ NOx ppb level (Table S3). After α values are calculated for a specific HC ppbC/ NOx ppb level, they can be scaled between fresh and aged compositions as described below. By doing so, the UNIPAR model is able to leverage the complexity of a relatively large semi-explicit gas oxidation mechanism (generally between 200 to 500 non-radical species per precursor from MCM) while limiting the computational load. In the model, autoxidation products are typically allocated to low volatility, and low reactivity groups (volatility group 1 in reactivity groups P and S).

Atmospheric aging can augment the product distribution, forming more reactive and less volatile products via oxidation, or photolysis products which are more volatile but may be more reactive. The stoichiometric coefficients of the $\alpha_i$ array are dynamically predicted as a function of $NO_x$ and aging as described in previous literature (Zhou et al., 2019; Han and Jang, 2020; Yu et al., 2021). A weighted aging factor is used to dynamically change $\alpha_i$ values based on fresh and highly oxidized compositions. The aging factor ($f_a$) at a time $t$, as detailed in Zhou et al. (2019), is defined as:

$$f_a(t) = \log \frac{[HO_2] + [RO_2]}{[HC]_0} \tag{1}$$

where $[HO_2]$ and $[RO_2]$ and $[HC]_0$ are the concentrations of hydroperoxide radical, organic peroxyl radical, and initial HC, respectively. Generally, the amount of oxidation within a given system is correlated with the concentrations radicals within the system, with higher concentrations of radicals in more aged systems. Thus, concentrations of major radicals, normalized by initial hydrocarbon concentration, are used to represent the amount of aging. Both fresh and highly oxidized $\alpha_i$ arrays are constructed for each $NO_x$ level as well as the respective aging factors. $f_a(t)$ is also used to generate an aging scale that ranges from 0 (fresh composition) to 1 (highly oxidized composition) as follows (Zhou et al., 2019):

$$f_a'(t) = \frac{f_a(highly\ oxidized) - f_a(t)}{f_a(highly\ oxidized) - f_a(fresh)} \qquad (2)$$

$f_a'(t)$ is calculated for a given $NO_x$ level and used to dynamically calculate the $\alpha_i$ values for that same $NO_x$ level as follows (Zhou et al., 2019):

$$\alpha_i = \big(1 - f_a'(t)\big)(fresh\ \alpha_i) + \big(f_a'(t)\big)(highly\ oxidized\ \alpha_i) \qquad (3)$$

Essentially, $[HO_2]$ and $[RO_2]$ are used to scale the oxidation product distribution of a given HC precursor (i.e. $\alpha_i$ array) between a fresh and aged composition. Physicochemical parameters of lumping species include MW, O:C ratio, and HB and are used to calculate multiphase partitioning and in-particle chemistry in the UNIPAR model. Further details about lumping criteria and physicochemical parameters can be found in the study by Zhou et al. (2019).

### 3.4 Construction of branched alkane lumping arrays

In the current UNIPAR model, the lumping arrays of a variety of linear alkanes can be predicted using the IVC approach (Madhu et al., 2023). To extend the lumping array of linear alkane to branched alkane, the volatility drop caused by branched alkyl groups was considered by matching each branched alkane to the linear alkane with the nearest vapor pressure. In addition, the lumping array in the model simulates the impact of the degree and position of branched alkyl groups on reducing the ability of branched alkanes to undergo autoxidation reactions. Figure S2 illustrates the ability of branched alkyl groups to reduce the ability of branched alkanes to undergo autoxidation reactions. For example, n-heptadecane has fourteen secondary carbons which can undergo autoxidation mechanism as seen in Fig. S2 (Left), while 2,2,4,4,6,8,8-heptamethylnonane has only one secondary carbon which can process autoxidation due to the seven methyl branches which reduce the number of hydrogens available for abstraction. To account for this, an autoxidation reduction factor (ARF) is applied to the lumping array of branched alkane and calculated as follows:

$$ARF = \frac{AP\ of\ branched\ alkane}{AP\ of\ linear\ alkane\ with\ nearest\ vapor\ pressure} \qquad (4)$$

where the autoxidation potential (AP) can be calculated as follows:
a) Terminal, primary, carbons are not included for AP.
b) Carbons that are in the $\alpha$ or $\beta$ position relative to terminal carbons on the alkane backbone are assigned an AP value of 1 and this carbon can only undergo autoxidation reactions in one direction. For example, this would apply to carbons 2, 3, 14, and 15 in Fig. S2A.
c) Other carbons on the main alkane backbone are assigned an AP value of 2 due to the potential for autoxidation occurring in two directions. For example, this would apply to carbons 4 to 13 in Fig. S2A.
d) If 1 alkyl branch is present on a carbon, then the AP value is multiplied by 0.5.
e) If geminal alkyl branches are present on a carbon, then the AP value is multiplied by 0.
f) AP value of each alkane is the sum of all AP values for each carbon.

Further information on the basis for the calculation of the ARF can be found in Section S3. ARF is Table S1 summarizes the physical properties and ARF values for branched alkanes used in the chamber experiments of this study. Physicochemical parameters, and α-values of lumping arrays of compounds used for chamber experiments can be found in Sections S5 and S6. Ultimately, this approach allows for the reduction of SOA formation found in branched alkanes compared to that in linear alkanes with same carbon numbers.

## 3.5 SOA formation by multiphase partitioning

The partitioning process of lumping species is fundamental to form SOA and process in-particle chemistry. Partitioning coefficients for each lumping species, $i$, between $g$ and $org$ phase ($K_{or,i}$) or between the $g$ and wet $inorg$ phase ($K_{in,i}$) are calculated using the typical gas-particle partitioning model (Pankow, 1994):

$$K_{or,i} = \frac{7.501RT}{10^9 MW_{org}\, \gamma_{org,i} p_{L,i}^o} \tag{5}$$

$$K_{in,i} = \frac{7.501RT}{10^9 MW_{inorg}\gamma_{inorg,i} p_{L,i}^o} \tag{6}$$

where R is the gas constant (8.314 J mol$^{-1}$ K$^{-1}$), and T is temperature (K). $MW_{org}$ and $MW_{inorg}$ are the average molecular weights (g mol$^{-1}$) of the organic and inorganic phases of the aerosol, respectively. $p_{L,i}^o$ is the subcooled liquid vapor pressure of a species, $i$. The activity coefficient in $or$ phase for each lumping species, $\gamma_{org,i}$, is assumed to be unity (Jang and Kamens, 1998). The activity coefficient in $inorg$ phase for each lumping species, $\gamma_{inorg,i}$, is predicted by a semi-empirical regression equation which was fit to the activity coefficients of various organic compounds as a function of physicochemical parameters (MW, O:C ratio, and HB) and sulfate fraction (FS). FS is an indicator for aerosol acidity which is defined as follows:

$$FS = \frac{[SO_4^{2-}]}{[SO_4^{2-}] + [NH_4^+]} \tag{7}$$

where $[SO_4^{2-}]$ and $[NH_4^+]$ are the concentration of sulfate and ammonium ions, respectively. The semi-empirical equation, derived from activity coefficients, estimated using the Aerosol Inorganic-Organic Mixtures Functional Groups Activity Coefficients (AIOMFAC) model (Zuend et al., 2011) at a given RH, is as follows:

$$\gamma_{inorg,i} = e^{0.035 \cdot MW_i - 2.704 \cdot \ln(O:C_i) - 1.121 \cdot HB_i - 0.330 \cdot FS - 0.022 \cdot (\cdot RH)} \tag{8}$$

Further information on the derivation and statistical properties of Eq. (8) can be found in Zhou et al. (2019). The partitioning coefficients are used to calculate the concentration of each lumping species in the three phases ($C_{g,i}$, $C_{org,i}$, and $C_{inorg,i}$) from the total concentration of each lumping species ($C_{T,i}$). The total SOA mass formed by partitioning ($OM_P$) in both $org$ and $inorg$ phases is predicted by the following equation which was developed by Schell et al. (2001) and reconstructed to consider mass formed by particle-phase reactions ($OM_{AR}$), seen in section 3.6, by Cao and Jang (2010):

$$OM_P = \sum_i \left[ C_{T,i} - OM_{AR,i} - C_{g,i}^* \frac{\frac{C_{org,i}}{MW_i}}{\sum_i \left(\frac{C_{org,i}}{MW_i} + \frac{OM_{AR,i}}{MW_{oli,i}}\right) + \frac{OM_0}{MW_{oli,i}}} \right] \tag{9}$$

where $Cg^*$ $(1/K_{org,i})$ and $OM_0$ (mol m$^{-3}$) represent the effective saturation concentration and pre-existing OM, respectively. $MW_{oli,i}$ and $MW_i$ represent the molecular weights of oligomeric products and lumping species, respectively. Eq. (9) is solved

using Newton-Rapson method, which iterates until a convergence is reached (Press et al., 1992).

## 3.6 SOA formation by particle-phase reactions

$OM_{AR}$ is formed in both the *org* and *inorg* phases. The inclusion of particle-phase reactions has been demonstrated to significantly improve predictions for aromatic hydrocarbons (Im et al., 2014; Zhou et al., 2019). Particle-phase reactions were also included in the study by Madhu et al. (2023) which demonstrated a negligible impact on linear alkane SOA. In *org* phase,

SOA formation is attributed to oligomerization as organic species undergo self-dimerization reactions (Han and Jang, 2020; Im et al., 2014; Yu et al., 2021; Zhou et al., 2019). In *inorg* phase, oligomerization of organic species can be accelerated by an acid catalyst (Jang et al., 2002). Oligomerization is expressed as a 2$^{nd}$-order reaction (Odian, 2004) with rate constants $k_{AR,org,i}$ and $k_{AR,inorg,i}$ (L mol$^{-1}$ s$^{-1}$) in *org* and *inorg* phases, respectively. $k_{AR,inorg,i}$ is described as follows:

$$k_{AR,inorg,i} = 10^{0.25pK_{BH^+_i}+1.0X+0.95R_i+\log(a_w[H^+])-2.58}$$ (10)

where $R_i$ represents species reactivity, $pK_{BH^+_i}$ represents the protonation equilibrium constant, $a_w$ represents the activity of water, $X$ represents excess acidity (Cox and Yates, 1979), and $[H^+]$ represents the concentration of protons which are estimated using the extended aerosol inorganic model (E-AIM (Clegg et al., 1998)) $k_{AR,org,i}$ is described as follows:

$$k_{AR,org,i} = 10^{\left[0.25pK_{BH^+_i}+0.95R_i+1.2\left(1-\frac{1}{1+e^{0.005(300-MW_{org})}}\right)+\frac{2.2}{1+e^{6(0.75-O:C)}}-10.07\right]}$$ (11)

For the oligomerization in *org* phase, the terms related to acidity ($X$, and $a_w[H^+]$) are excluded. As explained by Zhou et al.

(2019), a significant uncertainty remains in the calculation of $[H^+]$ specifically in low RH and ammonia rich environments due to a poor performance of the E-AIM under these conditions (Li and Jang, 2012). Studies have previously demonstrated that aerosol viscosity can influence the mobility of chemical species and thus, apparent reaction rates, which can be limited by slow bulk diffusion in the particle-phase (De Schrijver and Smets, 1966; Reid et al., 2018). The molecular weight of species in the organic phase ($MW_{org}$) and the O:C ratio, which are important predictors for viscosity, are considered to calculate $k_{AR,org,i}$.

Han and Jang (2022) demonstrate this method which accounts for viscosity in their application to SOA predictions from gasoline vapor composed of aromatic hydrocarbons and long-chain alkanes. Sulfuric acid can react with reactive organic compounds in the wet *inorg* phase of the aerosol to form dialkyl sulfate (diOS). The formed diOS can contribute to SOA mass production and leads to a reduction in [H$^+$] which decreases the rate of SOA mass produced by acid-catalyzed oligomerization in *inorg* phase. The formation of diOS is simulated in the UNIPAR model and reduces [H$^+$] in *inorg* phase as previously

reported (Im et al., 2014; Beardsley and Jang, 2016; Zhou et al., 2019).

## 3.7 Correction of intermediate organic vapor deposition to walls

Semi-volatile oxidized products derived from precursor HCs can deposit to chamber walls. As described in the previous studies by Han and Jang (2020), and Han and Jang (2022) the organic vapor deposition to wall is kinetically treated at the given chamber with the deposition ($k_{on,i}$) and desorption ($k_{off,i}$) rate constants of each lumping species, $i$. $k_{on,i}$ is expressed as a fractional loss rate (Mcmurry and Grosjean, 1985):

$$k_{on,i} = \left(\frac{A}{V}\right) \frac{\alpha_{w,i}\bar{v}_i/4}{1 + \frac{\pi\alpha_{w,i}\bar{v}_i}{8(K_e D)^{1/2}}} \tag{12}$$

where $D$ ($1.0 \times 10^{-6}$ m$^2$ s$^{-1}$) and $K_e$ (0.12 s$^{-1}$) are the diffusion coefficient and coefficient of eddy diffusion applied as a fixed number, respectively. $\left(\frac{A}{V}\right)$ represents the surface area to volume ratio of the chamber. $\bar{v}_i$ and $\alpha_{w,i}$ represent the mean thermal speed of the gas molecules, and accommodation coefficient of $i$ to the wall, respectively. Further information regarding the calculation of $\bar{v}_i$ and $\alpha_{w,i}$ can be found a previous study (Madhu et al., 2023). $K_{w,i}$ ($K_{w,i} = k_{on,i}/k_{off,i}$) is calculated as follows:

$$\ln(K_{w,i}) = -\ln(\gamma_{w,i}) - \ln(p^o_{L,i}) + \ln\left(\frac{7.501 RT OM_{wall}}{10^9 MW_{OM}}\right) \tag{13}$$

$OM_{wall}$ (mg m$^{-3}$) and $MW_{OM}$ are the concentration of organic matter on the wall, and the molecular weight of organic matter on the wall, respectively. The activity coefficient ($\gamma_{w,i}$) of lumping species, $i$, in $OM_{wall}$ is calculated using the quantitative structure-activity relationship (QSAR) approach with the physicochemical properties $H_{d,i}$, $H_{a,i}$, and $P_i$ which represent hydrogen bond acidity, hydrogen bond basicity, and polarizability of each lumping group $i$, respectively (Abraham et al., 1991; Abraham and Mcgowan, 1987; Leahy et al., 1992; Platts et al., 1999; Puzyn et al., 2010). Eq. (13) can be rewritten as:

$$\ln(K_{w,i}) = -(a_p H_{d,i} + b_p H_{a,i} + r_p P_i + c_p) - \ln(p^o_{L,i}) + \ln\left(\frac{7.501 RT OM_{wall}}{10^9 MW_{OM}}\right) \tag{14}$$

The values of $H_{d,i}$, $H_{a,i}$, and $P_i$ are calculated with the PaDEL-Descriptor, (Yap, 2011). The value of $K_{w,i}$ is used along with the $k_{on,i}$ to predict lumping species' wall loss using an analytical equation from the study by Han and Jang (2020) as follows:

$$C_{g,i} = \frac{K_{w,i} C_{T,i}}{K_{w,i}+1} e^{-k_{on,i}\left(1+\frac{1}{K_{w,i}}\right)t} + \frac{C_{T,i}}{K_{w,i}+1} \tag{15}$$

where $C_{g,i}$ (µg m$^{-3}$) is the gas-phase concentration of a lumping species, $i$, after time step $t$ (360 s). $C_{T,i}$ (µg m$^{-3}$) is the sum of $C_{g,i}$ and the concentration of lumping species $i$ on the chamber wall ($C_{w,i}$ (µg m$^{-3}$)). This method for correcting the bias originating from gas-wall partitioning has been previously demonstrated for toluene, TMB, α-pinene (Hang and Jang, 2020), and linear alkanes (Madhu et al., 2023), as well as a composition of gasoline vapor (Han and Jang, 2022). As explained by Hang and Jang (2020), uncertainties in this method are associated with the calculation of physicochemical parameters of lumped groups. The properties of gas-wall partitioning for branched alkanes were inherited from linear alkanes.

### 3.8 UNIPAR procedure for SOA mass production each time step

At each step, $C_{T,i}$ is estimated by using the newly produced $\Delta HC$ and $\alpha_i$, and it is combined with the previous step's concentration of lumping species, except those used for the formation of $OM_{AR}$ and organic vapor deposition to walls for the simulation of chamber data. Then, the updated $C_{T,i}$ is applied to generate $C_{g,i}$, $C_{org,i}$, and $C_{inorg,i}$ based on multiphase partitioning coefficients as seen in Eq. (5) and Eq. (6). $C_{inorg}$ and $C_{org}$ are then used to form $OM_{AR}$ via oligomerization in both the *inorg* and *org* phases with the rate constants calculated in Eq. (10) and Eq. (11), respectively. In the model, the quantity of the sulfate associated with OS in the *inorg* phase is also estimated and applied to recalculate [H$^+$]. After the process to form $OM_{AR}$, the remaining concentration of lumping species is used to estimate the organic vapor deposition to the wall using Eq. (15). $OM_P$ is calculated using a Newtonian approach (Eq. 9) in the presence of $OM_{AR}$ and the preexisting $OM_0$ at the end of each time step. For the total SOA mass, $OM_{AR}$, $OM_P$ and $OM_0$ are combined.

## 4 Results and Discussion

### 4.1 Chamber data vs. model prediction

The feasibility of the UNIPAR model to predict the SOA formation from various branched alkanes (Table S1) was demonstrated by comparing simulations with chamber data collected under various experimental conditions in the UF-APHOR chamber. As seen in Fig. 2, the UNIPAR model is reasonably able to predict the SOA formation from 2,6,10-Trimethyldodecane (C15), 2,2,4,4,6,8,8-Heptamethylnonane (C16), and 2,6,10,14-Tetramethylpentadecane (C19) at both high and low NO$_x$ conditions under different seed conditions (Table 1). Gas simulations used to predict HC consumption, RO$_2$ concentrations, and HO$_2$ concentrations performed using the CB6 mechanism can be seen in Fig. S3. As seen in the gas simulation, consumption of Isododecane was overpredicted compared to chamber measurements. This is likely due to the relatively low purity (80%) of commercially available Isododecane that was used for the chamber experiment. However, even with an overprediction for the gas consumption, the UNIPAR model SOA showed only a slight overprediction, indicating that the relatively low SOA yield of Isododecane is well represented within the model.

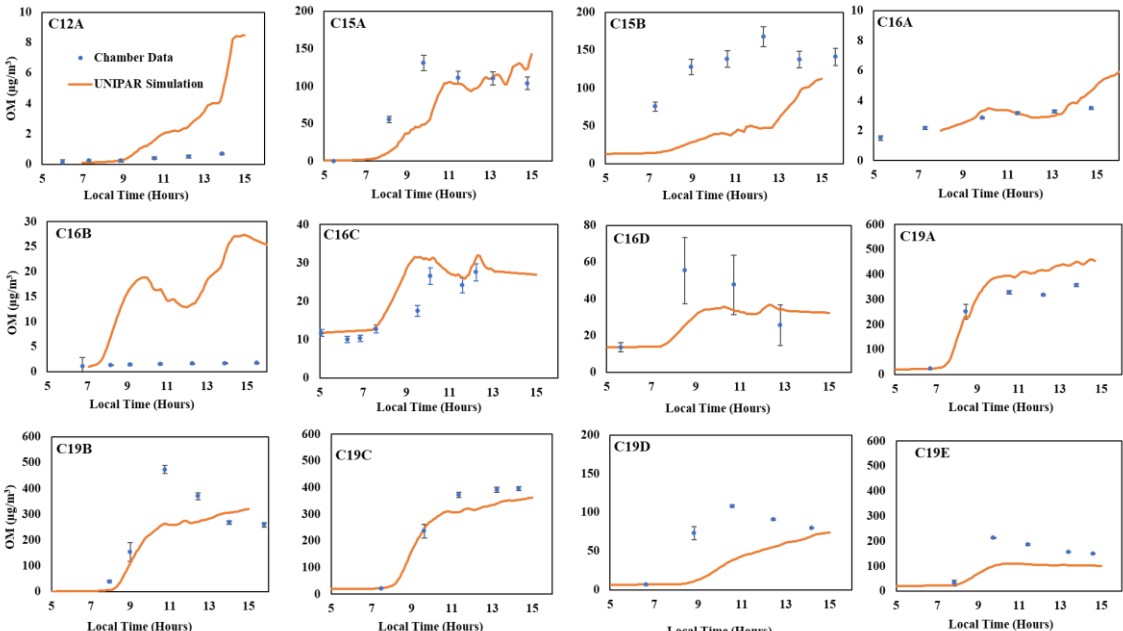

Figure 2. Comparison of SOA mass produced between simulations for Isododecane, Trimethyldodecane, Heptamethylnonane, and Tetramethylpentadecane to chamber data (Table 1). The blue dots represent observed SOA data collected that are corrected for particle wall loss to the chamber. Error bars represent a 95% confidence interval for each data point.

Generally, the presence of branching significantly reduced the amount of alkane SOA mass. This impact can be seen when comparing chamber data generated for Isododecane (branched C12) to linear C12 and chamber data generated for Trimethyldodecane (branched C15) to linear C15 as seen in Fig.3. Remarkably, highly branched C16 (Fig. 2) shows a lower capability for SOA formation compared to linear C15 (Fig. 3). Overall, the typical impact of $NO_x$ levels on SOA formation appeared, showing a negative relationship. Unlike SOA generated from aromatics (Im et al., 2014), branched alkane SOA was

insensitive to seed condition due to the low polarity of the products. A similar result is observed in linear alkane SOA reported by Madhu et al. (2023). Further discussion on the impact of alkyl branches, $NO_x$ conditions, humidity, seed, and temperature can be found in the upcoming sections 4.3 and 4.4.

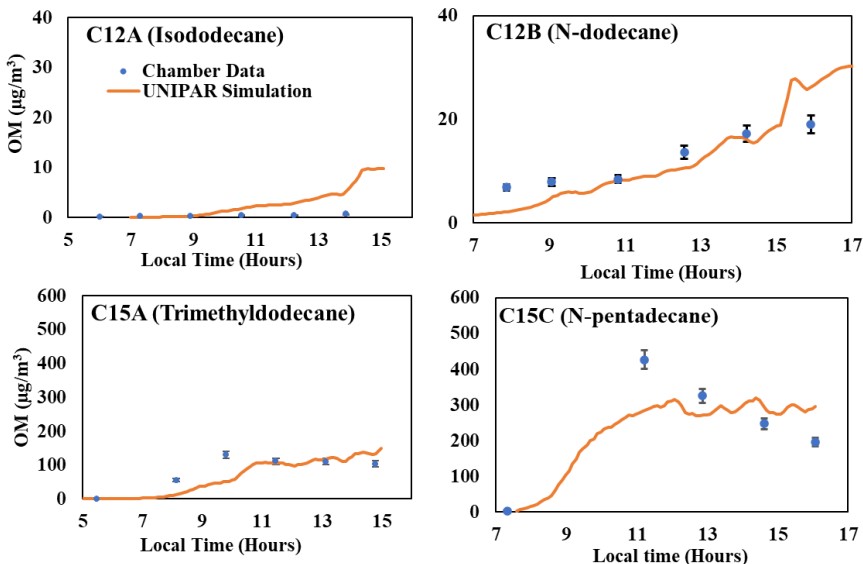

Figure 3. Comparison of the SOA mass produced by two branched alkanes (Isododecane and Trimethyldodecane) to the mass produced by linear alkanes (Madhu et al., 2023) of the same carbon number (Table 1). Error bars for C12A and C15A represent a 95% confidence interval for each data point. Error bars for C12B and C15C are 8% as reported by Madhu et al. (2023).

## 4.2 Characterization of aerosol composition

Figure 4 displays the relative functional group compositions of various alkane SOA constructed using FTIR data. FTIR spectra were decoupled into functional groups using the curve fitting method, assuming that a Gaussian distribution governs each peak. The decoupled FTIR bend for each functional group was applied to estimate the functional group composition of alkane aerosol using the relative intensity of the functional group determined from various reference compounds. The fitting parameters are the center frequency, the peak absorbance, and the half width at half-height. The relative functional group intensities for −OH, −COOH, C=O in ketones and aldehydes, C−O in non-alcohol and non-carboxylic acid groups, and NO3 in organonitrates were normalized with that of C−H stretching. The O:C ratios, calculated using functional group distributions from FTIR spectra, are also shown in Fig. 4 alongside model predicted O:C ratios. Values for peak ranges for each functional group and oxygen and carbon content values used to calculate O:C ratios can be found in Table S5. The model is able to reasonably predict O:C ratios for chamber generated data for both branched and linear alkanes. As expected, alkanes with a larger number of carbons tend to produce less oxidized SOA compared to alkanes with a smaller number of carbons. The relatively low O:C ratios found in alkane SOA systems support our assumption that organic and wet inorganic phases exist separately as most organic species are unlikely to be soluble in the inorganic phase (Yu et al., 2021a). When comparing the SA-seeded C19 and non-seeded C19 SOA systems, the SA system shows smaller amount of C=O functional groups but higher amount of C-O, evidently indicating

some acid-catalyzed oligomerization (Jang et al., 2002), although the impact of wet inorganic seed is small (section 4.4). All SOA systems shown in Fig. 4 are produced under relatively low $NO_x$ conditions (Table 1). Increasing the $NO_x$ levels during alkane oxidation may increase the amount of nitrate functional groups in SOA.

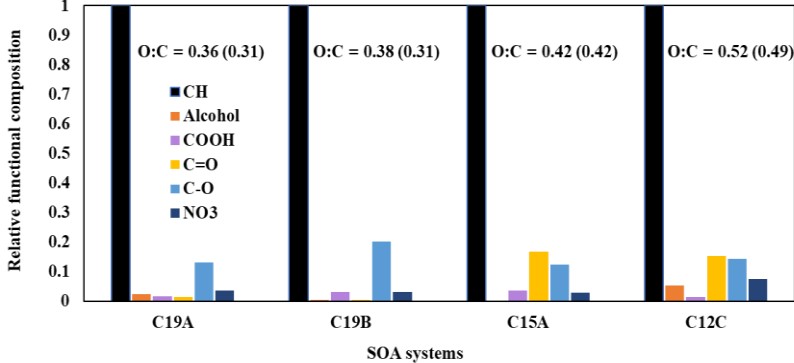

Figure 4. Relative functional group compositions of different SOA systems constructed using FTIR data. The O:C ratios of each SOA system, constructed with the functional group composition from FTIR data, are shown along with model-predicted O:C ratios displayed in parentheses.

## 4.3 Sensitivity of alkane SOA yields to alkyl branches

Figure 5 illustrates the impact of alkyl branches on the SOA Yields of an alkane with 15 carbons. Information on the structures, parameters used to generate lumping arrays, and calculated OH-radical reaction rates for alkanes used in Fig. 5 can be found in Table S5. Figure 5 clearly demonstrates the decrease in alkane SOA yields as the number of alkyl branches increases at given oxidation conditions. This result qualitatively agrees with previous literature which explore the impact of alkyl branches on SOA yields of alkanes (Loza et al., 2014; Lim and Ziemann, 2009; Tkacik et al., 2012). This figure also illustrates the relative importance of the ARF in the model compared to the vapor pressure drop due to branching. When the number of branches increases, the decrease in vapor pressure may not be large enough such that the linear alkane with the nearest vapor pressure changes. For example, C15 alkanes with one branch and two branches, at the given structures, both have vapor pressures nearest to that of linear C14 (Table S5). In this case, the ARF becomes the only method to decrease SOA yields and the decrease in SOA yields due to an increase in alkyl branches is relatively small within the model. This suggests that the primary driver in reduction of alkane SOA yields due to branching is the increase in vapor pressure, with the ARF being a significantly smaller component. Additionally, when comparing the C15 alkane with one branch to the one with two branches, the difference in SOA yields for the low $NO_x$ condition is slightly larger when compared to the difference in SOA yields for the high $NO_x$ condition. This indicates that, similarly to linear alkanes (Madhu et al., 2023), the fraction of branched alkane autoxidation products of total SOA mass in low $NO_x$ conditions is larger compared to that in high $NO_x$ conditions. Further discussion on the impact of $NO_x$ conditions on the SOA yields of branched alkanes can be found in the upcoming section 4.4.

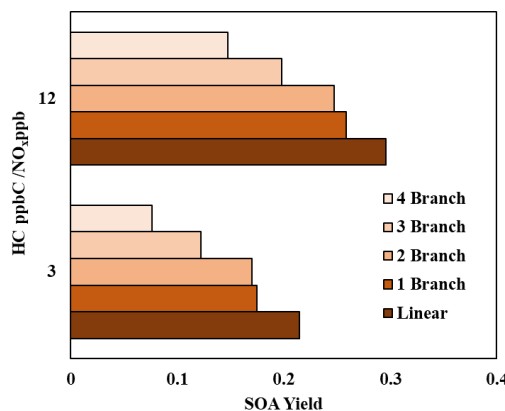

Figure 5. Simulated SOA Yields for 15 carbon alkanes with various number of branches at two different $NO_x$ levels. $OM_0= 5$ $\mu g/m^3$ , 298 K, RH = 60%, HC consumption = 100 $\mu g/m^3$, sunlight profile shown in Fig. S4.

## 4.4 Sensitivity of branched alkane SOA formation to $NO_x$ levels, temperature, humidity, and seed conditions

Figure 6 illustrates the SOA yields of three alkanes that each have 3 branched methyl groups (C12, C15, C18) under various $NO_x$ levels. Information on the structures, parameters used to generate lumping arrays, and calculated OH-radical reaction rates for alkanes used in Fig. 6 can be found in Table S6. Similarly to most SOA precursors, the simulated SOA yields of all three branched alkanes increased as $NO_x$ levels decreased. Because gas simulations are conducted in a manner which the HC consumption is kept fixed for each condition, the difference between SOA yields at different $NO_x$ levels can be attributed to a change in product distributions. In high $NO_x$ conditions, the paths to form organonitrate products can compete with the paths which form low-volatility products via autoxidation. In addition, the formation of organonitrate suppresses the further oxidation of products. Less oxidized products tend to be less volatile products, which reduce SOA yields. This result is in agreement with  Loza et al. (2014) who find a higher yield for Isododecane under low $NO_x$ conditions when similar amounts of hydrocarbon are consumed in both high and low $NO_x$ conditions.

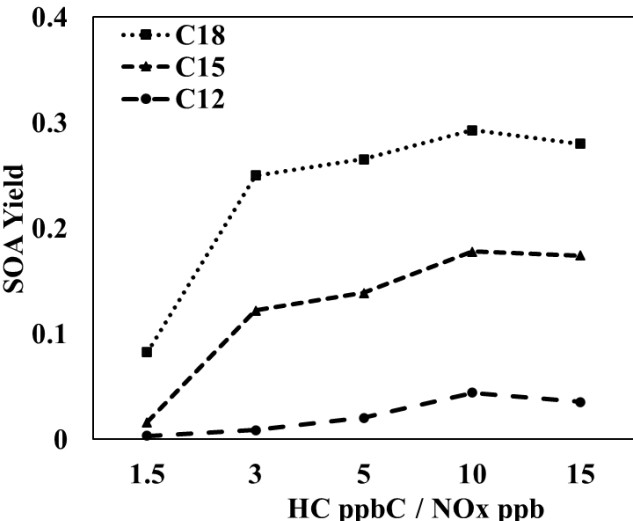

Figure 6. Simulated SOA yields for three-branched alkanes of different carbon numbers (C12, C15, C18) at various HC ppbC/NO$_x$ ppb levels. OM$_0$= 5 μg/m$^3$, 298 K, RH = 60%, HC consumption = 100 μg/m$^3$, sunlight profile shown in Fig. S4.

Figure 7 illustrates the SOA yields of 2,6,10-Trimethyldodecane under 3 different seed conditions (ammonium sulfate (AS), sulfuric acid (SA), and no seed (NS)) and two RH conditions (30% and 60%). Under 60% RH, the AS seed is wet, and under 30% RH, the AS seed is dry. Similarly to linear alkanes (Madhu et al. 2023), branched alkanes SOA yields were not significantly impacted by the presence of seed under either humidity condition. Additionally, no significant impact of acidic seed on branched alkane appeared, indicating that alkane SOA formation is dominated by partitioning rather than particle phase

chemistry. This is consistent with the FTIR data (Fig. 4) which shows a lack of reactive aldehydic C=O species which can be involved in acid-catalyzed oligomerization.

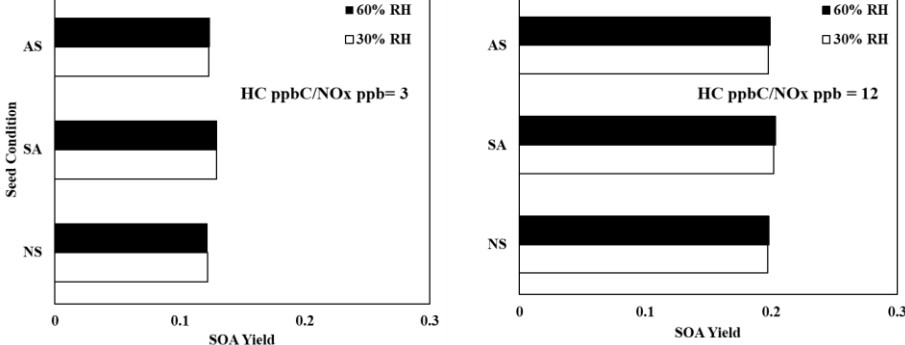

Figure 7. Simulated SOA Yields for 2,6,10-Trimethyldodecane at various seed conditions (10 μg/m$^3$ Ammonium Sulfate (AS), 10 μg/m$^3$ Sulfuric Acid (SA), and no seed (NS)) and 2 HC ppbC / NOx ppb levels. OM$_0$= 5 μg/m$^3$, 298 K, RH = 60%, HC

consumption = 100 μg/m$^3$, sunlight profile shown in Fig. S4.

Figure 8 displays the SOA yields of three different 3-branched alkanes (Table S6) under 3 different temperatures and two different NO$_x$ conditions. As expected, due to the relatively high importance of partitioning, the SOA yields of all the 3-branched alkanes are significantly impacted by changes in temperature under both NO$_x$ conditions. Additionally, the impact of temperature on SOA yields decreases as the number of carbons increase because larger molecules typically produce more low-volatility products which tend to exist in the particle-phase at various temperatures.

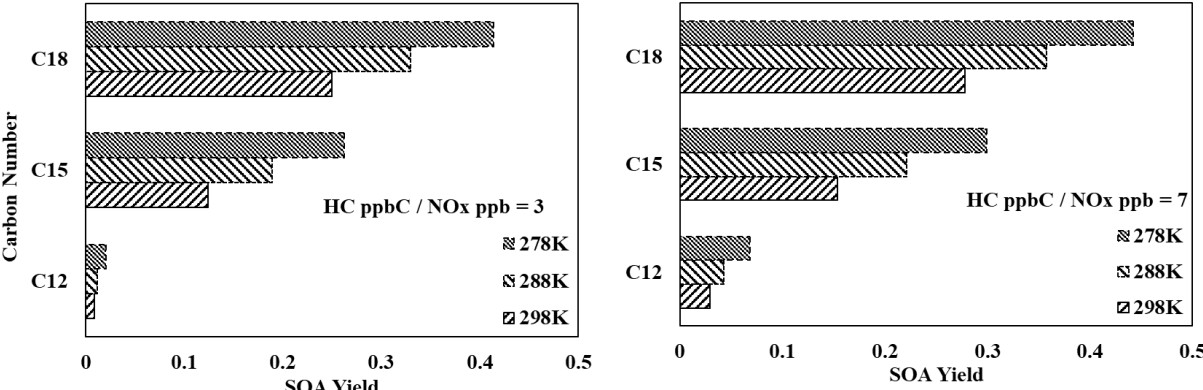

Figure 8. Simulated SOA Yields from photooxidation of three different 3-branched alkanes (Table S3) at three different temperatures (278K, 288K, 298K) and two different NO$_x$ levels (HC ppbC/NO$_x$ ppb = 3, 7). OM$_0$= 5 µg/m$^3$, RH = 60%, HC consumption = 100 µg/m$^3$, sunlight profile shown in Fig. S4.

## 4.5 Uncertainty of model rate constants

Figure 9 illustrates the impact of increasing and decreasing the UNIPAR oligomerization rate constant by a factor of 2 on the SOA yields of 2,6,10-Trimethyldeodecane at both high and low NO$_x$ levels. Unlike aromatic SOA (Im et al., 2014; Zhou et al., 2019; Han and Jang, 2022), the SOA yields were not significantly impacted by either change to the oligomerization rate constant under both NO$_x$ conditions. This reaffirms the previously discussed (Sections 4.2, 4.4) idea that particle-phase reactions (oligomerization reactions) serve a relatively small part in branched alkane SOA formation due to the lack of reactive oxidation products. Thus, within this model, there is a low level of uncertainty which originates from the rate constant for particle phase reactions.

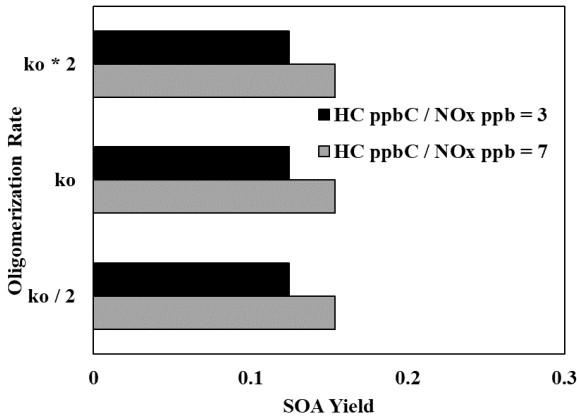

Figure 9. Impact of increasing and decreasing the UNIPAR oligomerization rate constant by a factor of 2 on the SOA yields of 2,6,10-Trimethyldodecane at two different $NO_x$ levels (HC ppbC / $NO_x$ ppb = 3, 7). $OM_0$= 5 μg/m$^3$, RH = 60%, HC consumption = 100 μg/m$^3$, sunlight profile shown in Fig. S4.

Vapor pressures related to volatility groups were calculated using a group contribution method that has an estimated uncertainty of a factor of 1.45 (Zhao et al., 1999; Myrdal and Yalkowsky, 1997). Fig. 10 displays the impact of uncertainties in the estimation of vapor pressure in UNIPAR. Contrary to the oligomerization rate constant, changing the lumping group vapor pressures causes significant changes in SOA yield which demonstrates the important role of partitioning in branched alkane SOA formation.

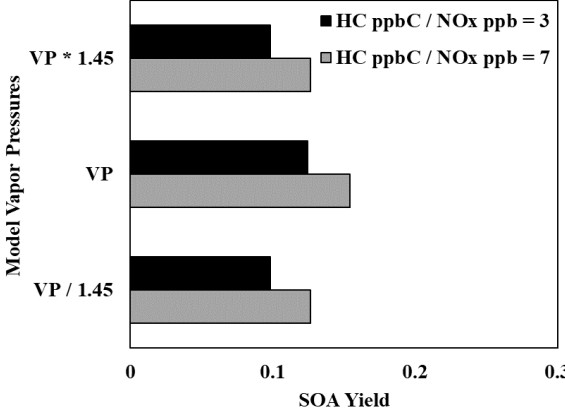

Figure 10. Impact of increasing/decreasing the UNIPAR lumping group vapor pressures by a factor of 1.45 on the SOA yields of 2,6,10-Trimethyldodecane at two different $NO_x$ levels (HC ppbC / $NO_x$ ppb = 3, 7). $OM_0$= 5 μg/m$^3$, RH = 60%, HC consumption = 100 μg/m$^3$, sunlight profile shown in Fig. S4.

**4.6 Application of IVC-base product distributions to SOA simulation from diesel linear and branched alkanes**

Diesel fuel is comprised of various linear and branched alkanes dominantly ranging from C9 to C24. The composition of linear and branched alkanes in diesel fuel, reported by Gentner et al. (2012), was applied to the UNIPAR model in Fig. 11. The gas simulation of diesel fuel was performed using the CB6 Ozone mechanism with relative concentrations of diesel fuel linear and branched alkanes (100 μg/m$^3$ total), as well as other common diesel constituents, as reported by Sazhin et al. (2014), under urban conditions (high NO$_x$ level). SOA formation was simulated only from branched and linear alkanes but the inclusion other diesel constituents allows for more accurate predictions of concentrations RO$_2$ and HO$_2$, as well as individual HC consumptions. It is important to note that the composition of branched and linear alkanes in fuels will vary from the composition of those in fuel exhausts. However, literature which reports on the composition of fuel exhaust typically reports significant proportions of branched alkanes as unspeciated branched alkanes (Lu et al., 2018; Tkacik et al., 2014). Thus, current diesel exhaust compositions cannot be used for this analysis. Each branched alkane was assumed to have 3 methyl branches. Within the UNIPAR model, SOA formation from all linear and branched alkanes was performed simultaneously, such that the SOA mass formed from one precursor can enhance the SOA mass formed from every other precursor. As seen in Fig.11, branched alkanes represent a higher proportion of diesel fuel and also SOA mass formed compared to linear alkanes. Branched alkanes represented 78% of the alkane HC input and were responsible for 72% of the total SOA mass produced.

Additionally, long-chain alkanes ($\geq$ C15) are relatively more important for SOA formation compared to smaller alkanes within both the linear and branched subsets. Long-chain linear alkanes represented 59% of the linear alkane composition in diesel and was responsible for 73% of the total linear alkane SOA mass production. Similarly, long-chain branched alkanes represented 56% of the branched alkane composition in diesel fuel and was responsible for 75% of the total branched alkane SOA mass production. Our conclusion regarding the importance of long-chain alkanes generally agrees with the conclusion presented by Madhu et al. (2023). However, the linear alkane SOA formation simulation by Madhu et al. (2023) was performed individually for each precursor, which did not allow for SOA mass produced from one precursor to influence the others. Thus, Fig. 11 is a better representation of the SOA formation potential from linear and branched alkanes in diesel fuel. However, the inclusion of other diesel constituents (e.g., cyclic alkanes, polyaromatics HCs, and aromatics) may further augment the SOA formation potential of the various linear and branched alkanes diesel fuel (Gentner et al., 2012).

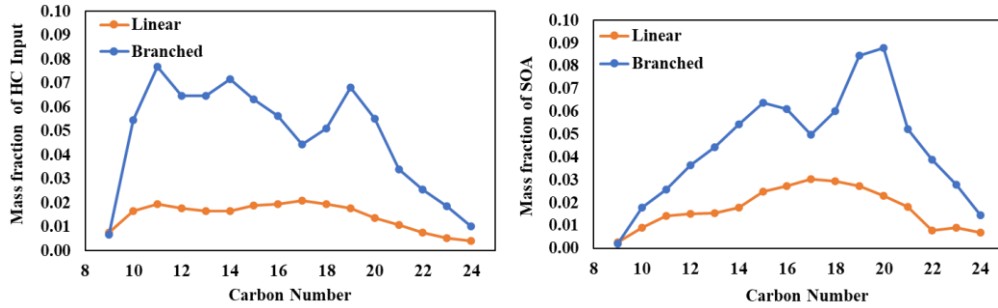

Figure 11. SOA formation from the photooxidation of diesel fuel linear and branched alkanes in the presence of NO$_x$. Composition as reported by Gentner et al. (2012). Concentration of initial HC = 993 ppbC, temperature = 298 K, RH = 60%, and HC ppbC/ NOx ppb= 3, sunlight profile shown in Fig. S4.

## 4.7 Summary and Conclusions

Branched alkanes are one of the major classes of HCs in urban environments, specifically representing significant proportions of both gasoline and diesel fuel (Gentner et al., 2012). This study models SOA formation using the UNIPAR model via the multiphase reactions of branched alkanes. Due to a lack of practically applicable gas mechanisms available for a variety of branched alkanes, the lumping arrays of branched alkanes were predicted using previously existing lumping arrays of linear alkanes. To do so, the lumping array of each branched alkane was primarily created using the lumping array of the linear alkane with the nearest vapor pressure. In addition to a decrease in vapor pressure, branching present on an alkane chain can reduce the ability of the oxidation products to undergo autoxidation (Fig. S2). Autoxidation has been demonstrated in previous studies which show that autoxidation products significantly contribute to terpene SOA (Pye et al., 2019; Xavier et al., 2019; Yu et al., 2021), and linear alkane SOA (Madhu et al., 2023). To account for the reduction of autoxidation products in branched alkane compared to that in linear alkanes, an ARF value (Eq. 4) was applied to the α-values of lumping groups. Lumping arrays generated in this manner were applied within UNIPAR to predict SOA formation from branched alkanes, which was compared to chamber data. Notably, the presence of alkyl branches can also significantly increase the amount of decomposition reactions which produce more volatile products (Loza et al., 2014; Lim and Ziemann, 2009; Tkacik et al., 2012). Whereas the increase in decomposition reactions due to alkyl branches is not explicitly accounted for, our application of the ARF implicitly captures some increase in decomposition. We apply the ARF to reduce the value of stoichiometric coefficient related to the lumped groups which yield autoxidation products. The amount of stoichiometric coefficient which is expelled from the α$_i$ array via this reduction is essentially treated as products which are so highly volatile that they cannot form SOA via gas-particle partitioning. The model predicted SOA formation well agreed with chamber data (Fig. 2). Additionally, O:C values of chamber generated SOA, which were calculated using FTIR spectra, also were in agreement with model predicted O:C values. Similarly to linear alkanes (Madhu et al., 2023), branched alkanes showed significant sensitivity to NO$_x$ levels as seen in Fig. 6. The degree of branching was also shown to significantly impact branched alkane SOA, with yields generally decreasing as the number of methyl branches increases (Fig. 5). As branching increases, the vapor pressure increase of the precursor and subsequent oxidation products was determined to a more significant factor contributing to branched alkane SOA yields than the ARF (section 4.3). The branched alkane SOA formation is dominated by gas-particle partitioning processes, particularly between the gas and organic phases due to the relatively non-polar, non-reactive oxidation products. Evidently, SOA yields are sensitive to temperature (Fig. 8), an environmental factor that is heavily tied to partitioning. Furthermore, branched alkane SOA yields are insensitive to changes in particle-phase reaction rates (Fig. 9) and show no significant impacts from changes in aerosol acidity or inorganic seed composition (Fig. 7).

The conclusions presented have several real-world implications. Firstly, branched alkanes are significant sources of SOA formation and should be considered as an SOA precursor, especially in urban environments where vehicular emissions represent a significant proportion of the emitted reactive organic carbon (Murphy et al., 2023). As shown in Fig. 11, branched alkanes within diesel were responsible for a significantly larger proportion of SOA mass production compared to linear alkanes. Secondly, the reduction of $NO_x$ concentrations in the atmosphere would not be effective to decrease branched alkane SOA formation as branched alkane SOA yields tend to increase as $NO_x$ levels decrease (Fig. 6). Thirdly, branched alkane SOA yields would not be significantly affected by the reduction of sulfate because of relatively non-polar, non-reactive oxidative products (Section 4.4). Notably, the chamber experiments that are used to validate the model results occur in half-day timescales. Further atmospheric aging could change oxidation product compositions and alter SOA mass yields. Additionally, compounds used in this study, which were picked from a limited set of commercially available branched alkanes, are generally larger and less branched compared to those identified in the atmosphere and in fuel exhausts. However, even in the case that the atmosphere contains significant amounts of small, highly branched compounds, those would be less relevant for SOA formation potential than compounds used in this study due to negligible SOA yields. Branched alkane SOA predictions using the UNIPAR model contain several sources of uncertainty. As previously explained, the UNIPAR parameters of each branched alkane were primarily inherited from an analogous linear alkane. For example, vapor pressures of precursor branched alkanes were matched with the linear alkane with the nearest vapor pressure. However, the two matched vapor pressures are rarely identical, and this deviation between the two values can yield uncertainty in the predicted lumping array. Similarly, physicochemical parameters (e.g. O:C, HB, and MW arrays, and wall loss parameters) inherited from linear alkanes can be included as sources of uncertainty in SOA prediction. Additionally, the branched alkane lumping arrays also inherited uncertainty associated with the linear alkane oxidation mechanisms that were originally used to generated linear alkane lumping arrays. As described by Madhu et al. (2023), the linear alkane gas oxidation mechanisms used to generate lumping arrays were written in such a way that, if a precursor has several possible points of reaction with a hydroxyl radical, only one path is included. The inclusion of alternative pathways may augment the linear alkane lumping arrays and have downstream effects on branched alkane lumping arrays. Uncertainties also exist in hydrocarbon consumption values simulated by CB6. As described in section S1, the CB6 model overpredicts hydroxyl radical concentrations for chamber experiments performed with long-chain alkanes. Accordingly, gas simulations had a tendency toward overprediction of hydrocarbon consumption. In this study, SOA formation from the photooxidation of linear and branched alkanes in diesel fuel was predicted by using UNIPAR model. However, diesel fuels also contain significant amounts of cyclic alkanes, which tend to have higher propensities to be SOA precursors (Manavi and Pandis, 2022; Loza et al., 2014; Lim and Ziemann, 2009). When considering SOA formation from alkanes in diesel, future studies should include cyclic alkanes to accurately predict SOA formation potentials.

**Code availability**

The code to run the UNIPAR model in this study is available upon request with appropriate purpose. The model parameters of UNIPAR are currently being updated to include more precursors, and the user manual is in preparation. When the manual and

parameters for essential precursors are ready, UNIPAR will be freely available for the public via GitHub. Chamber data and model simulations for branched alkane experiments performed for this study are publicly available on Zenodo (https://zenodo.org/records/10667797).

**Data availability**

The chamber data and simulated results used in this study are available upon request.

**Author contributions**

MJ designed the experiments, and AM and MJ carried them out. AM prepared the manuscript with contributions from MJ and
595 YJ.

**Competing interests**

The authors declare that they have no conflict of interest.

**Financial Support**

This research was supported by the National Institute of Environmental Research (NIER2021); the National Science Foundation (AGS1923651); and the Fine Particle Research Initiative in East Asia Considering National Differences (FRIEND) Project through the National Research Foundation of Korea (NRF) funded by the Ministry of Science and ICT (2020M3G1A1114556)

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
