# Peer review of "Modeling the influence of carbon branching structure on SOA formation via multiphase reactions of alkanes"

_EGUsphere, 2023_

## Author Comment (AC1)

**General Notes**

FTIR O:C calculations were updated. Within our O:C calculation, each C-H bond was counted as one carbon. However, one carbon can have multiple C-H bonds. A correction has been made to account for this discrepancy. The model still is in good agreement with O:C measurement.

To better reflect autoxidation literature, the calculation for the ARF has been updated and a further explanation of ARF calculation has been added to section S2. All table ARF values and model simulations have been updated. This change had negligible impacts on model simulations, and sensitivity and uncertainty tests.

**Response to Reviewer 1**

Thank you for reviewing this paper. We hope we have appropriately addressed all of your concerns below.

1) L219: It's unclear why branched alkanes have lower volatility compared to linear alkanes. I think methods for estimating vapor pressure like SIMPOL (Pankow and Asher, 2008) calculate the same value for both linear and branched alkanes. This is a crucial point throughout the paper and requires further justification.

*Response: Differences in vapor pressures between branched and linear alkanes cannot be determined by SIMPOL as this is a group contribution method that treats all carbons to be identical, regardless of if they are primary, secondary, or tertiary carbons. The group contribution method we use clearly differentiates between the vapor pressures of linear and branched alkanes with the same carbon number (Myrdal and Yalkowsky, 1997; Zhao et al., 1999; Stein and Brown, 1994).*

2) L329-330, Figure S2: When comparing C19B and C19C, the seed conditions are the only differing factors, yet large discrepancies exist in ozone simulation capability. What could be the underlying reason? Are there measurement errors or issues with the CB06 mechanism? A similar issue is observed between C16B and C16C, although in this case, initial NOx/HC concentrations also vary.

*Response: Incorrect chamber data for Ozone was displayed for C19B and has been updated. The discrepancy between the chamber measurement and gas simulation of Ozone has been reduced significantly after the figure was updated.*

3) L383-385: Alkanes with 1 and 2 branches show no significant difference. However, changes are distinct for others (0 vs 1, 2 vs 3, 3 vs 4). I initially thought ARF values for 1 and 2 would be similar, but Table S4 suggests otherwise. Could the authors explain this discrepancy?

*Response: As explained in L383-L385, the ARF is the only factor which accounts for the difference in yields for the C15 alkane with 1 branch and 2 branches. Thus, if the yields for the two compounds are similar, it suggests that the impact of the ARF is relatively small compared*

*to the reduction in vapor pressure due to branching. A sentence has been added for clarity and now reads:*

*(Section 4.3 Paragraph 1)"In this case, the ARF becomes the only method to decrease SOA yields and the decrease in SOA yields due to an increase in alkyl branches is relatively small within the model. This suggests that the primary driver in reduction of alkane SOA yields due to branching is the increase in vapor pressure, with the ARF being a significantly smaller component."*

4) L505: The results presented in this study are based on a timescale of less than half a day. In the real world, these compounds can undergo further aging. The authors should discuss this point when it comes to the real-world implications.

*Response: Some discussion has been added to address this point:*
*(Section 4.7, End of paragraph 2)"Notably, the chamber experiments that are used to validate the model results occur in half-day timescales. Further atmospheric aging could change oxidation product compositions and alter SOA mass yields."*

5) Data & Code Availability: Restricting access to data and code by simply stating they are "available upon request" seems inequitable, particularly in the year 2023. While it's understood that chamber data can be provided upon request, the rationale for not making the code publicly available is unclear. This lack of transparency is inconsistent with Copernicus's data policy (https://www.atmospheric-chemistry-and-physics.net/policies/data_policy.html#data_availability). I recommend utilizing free public repositories, such as Zenodo, for FAIR principles.

*Response: We are choosing to not publicly release the model code as of now as the model is still under development and parameters are being added for hydrocarbon species such as cyclic alkanes and wood smoke products such as phenols. Additionally, we are continuing to work on integrating the UNIPAR model into the CAMx regional model and are validating parameters via regional air quality simulations. The parameters for branched alkanes used in this study are reported in the SI. We are expecting to release the model as a whole with all relevant atmospheric species included in about 2 years.*

*Please find the information below in "Code availability" in the revised manuscript.*

**"Code availability**
*The code to run the UNIPAR model in this study is available upon request with appropriate purpose. The model parameters of UNIPAR are currently being updated to include more precursors, and the user manual is in preparation. When the manual and parameters for essential precursors are ready, UNIPAR will be freely available for the public via GitHub."*

**Minor comments.**

1) L46-47: Caplain et al. (2006) do not seem to discuss that branched alkanes represent larger proportions compared to linear alkanes. And I think it depends on the species. For example, n-butane is higher than isobutane but isopentane is higher than n-pentane.

*Response: The wrong paper was cited for this sentence. It should be Gentner et al. (2012) and this correction has been made. The concentration of linear alkanes in diesel fuel maybe larger than that of branched alkanes at some carbon numbers. However, if you sum all of the linear alkane mass in diesel fuel from C9-C24 (we did this manually using the relative heights of linear alkanes at each carbon number from their figure) and do the same for branched alkanes, then total amount of branched alkanes will be clearly larger than the linear alkanes. For example, we mention in section 4.6 that "Branched alkanes represented 78% of the alkane HC input", with the other 22% consisting of linear alkanes.*

2) L51: The proper name for the model is GECKO-A, not GECKOa.

*Response: This correction has been made.*

3) L106: Which protocol was used for OC/EC measurements? Thermal optical transmittance or thermal optical reflectance? OC/EC concentrations depend on the protocol, so some discussion is needed for their quantifications.

*Response: Data from thermal optical transmittance by using NDIR detector were used to measure OC. This information has been added into the experimental section (2):*

*"The OC/EC used a non-dispersive infrared detector (NDIR) which measured OC using thermal optical transmittance."*

4) L146-148: What about decomposition? Alkoxy radicals from branched alkanes could undergo decomposition, resulting in higher vapor pressures.

*Response: Decomposition is not explicitly accounted in this study. However, the ability of alkyl branched to induce decomposition of alkanes shows a similar tendency to the reduction of autoxidation caused by alkyl branches. We apply the ARF (see section 3.4) to reduce the value of stoichiometric coefficient related to the lumped groups which yield autoxidation products. The amount of stoichiometric coefficient which is expelled from the $\alpha_i$ array is essentially treated as products which are so highly volatile that they cannot form SOA via gas-particle partitioning. By doing this, we implicitly increase the amount of volatile oxidation products present within branched alkanes.*

*Please also find the discussion in the summary and conclusion section, which describes the potential limitation in the model.*

*(Section 4.7, Paragraph 1) "Notably, the presence of alkyl branches can also significantly increase the amount of decomposition reactions which produce more volatile products (Loza et al., 2014; Lim and Ziemann, 2009; Tkacik et al., 2012). Whereas the increase in decomposition reactions due to alkyl branches is not explicitly accounted for, our application of the ARF*

*implicitly captures some increase in decomposition. We apply the ARF to reduce the value of stoichiometric coefficient related to the lumped groups which yield autoxidation products. The amount of stoichiometric coefficient which is expelled from the αi array via this reduction is essentially treated as products which are so highly volatile that they cannot form SOA via gas-particle partitioning."*

5) L171: A reference for DSMACC is missing.

*Response: A citation has been added.*

6) L175: A detailed description of the CB6 mechanism is needed. There are several versions of CB6, such as CB6r1, CB6r2, CB6r3, and CB6r4. The authors may want to include a section describing the specific characteristics of the CB6 mechanism they used.

*A section describing the CB6 mechanism used for this study was added to the SI (Section S1).*

7) Table S1: Any reason for using Kwon and Atkinson, 1995, over a more recent framework like Jenkin et al. (2018)?

*Thank you for drawing our attention to this. Kwon and Atkinson is a very versatile framework that we have used historically. We have calculated rate constants using the Jenkin framework are they not significantly different for the alkanes we used for experiments as seen below.*

| Compound name | Reaction Rate with Hydroxyl Radical Kwok and Atkinson framework $(cm^3/(molecule*sec))$ | Reaction Rate with Hydroxyl Radical Jenkin framework $(cm^3/(molecule*sec))$ |
|---|---|---|
| Isododecane | 1.39 E-11 | 1.34 E-11 |
| 2,6,10-trimethyldodecane | 1.87 E-11 | 1.78 E-11 |
| 2,2,4,4,6,8,8-heptamethylnonane | 0.87 E-11 | 0.85 E-11 |
| 2,6,10,14-tetramethylpentadecane | 2.42 E-11 | 2.31 E-11 |

8) L370: Why does C12 show higher nitrate functional groups compared to C15 and C19? I think alkyl nitrate yield increases with increasing carbon numbers?

*Response: Alkyl nitrate yield does generally increase with increasing carbon number. However, C12C is an experiment performed with linear C12 whereas C15A is performed with 3-branched C15 (Trimethyldodecane) and C19A and C19B are performed with 4-branched C19 (Tetramethylpentadecane). The presence of branching alkyl groups promotes decomposition reactions which are competing with reactions that lead to the formation of organonitrate. Thus, Trimethyldodecane and tetramethylpentadecane may be less likely to form nitrate groups compared to linear C12.*

9) L400-401: While I agree with this point, how does the fate of RO2 change under high NOx conditions, particularly reactions like RO2 + HO2 vs RO2 + NO vs RO2 + RO2? I guess more RO2 + NO path under high-NOx conditions can enhance autooxidation?

*Response: Unfortunately, the impact of RO2 fates in relation to autoxidation of branched alkanes cannot be determined in this study without an explicit gas mechanism. However, Madhu et al. (2023) discusses the interaction between the autoxidation of linear alkanes and RO2 fate. We found that the autoxidation mechanism did not have any significant impact on RO2 fates. Additionally, autoxidation reactions occurred more quickly under low NOx conditions according to the integrated reaction rate analysis.*

10) L421-424: The logic is clear, but I can't see significant differences from Figure 8. To me, SOA changes due to temperature appear similar, regardless of NOx conditions.

*Response: Thank you for pointing this out. After close analysis, we find that the differences in yields due to temperature is very slightly higher in high NOx conditions compared to low NOx conditions. In the low NOx condition, C12 yields increased from by 57% as temperatures decreased from 298K to 278K. In the high NOx condition, C12 yields increased by 59% as temperatures decreased from 298K to 278K. Thus, this claim has been removed.*

11) L455: Genter et al. -> Gentner et al.

*Response: This correction has been made*

12) L462: Fig. 1 -> Fig. 11

*Response: This correction has been made.*

13) L501-502: The temperature dependency could vary significantly based on the enthalpy of vaporization parameter. What value was used in this study?

*Response: This information has been added to the text and now reads:*

*(Section 3.3, Paragraph 1)"The $\alpha_i$ array consists of 6 different reactivity levels (very fast (VF), fast (F), medium (M), slow (S), partitioning only (P), and multi-alcohol (MA)) and 8 different volatility levels (1E-08, 1E-06, 1E-05, 1E-04, 1E-03, 1E-02, 1E-01, and 1.0 mm Hg) based on vapor pressure which represent 48 species. The 8 volatility levels have enthalpy of vaporization*

*values 140E+3, 106E+3, 96E+3, 89E+3, 82E+3, 58E+3, 58E+3, and 58E+3 J/mol, respectively."*

**Response to Reviewer 1**

Thank you for reviewing this paper. We hope we have appropriately addressed all of your concerns below.

1) Introduction: My impression of UNIPAR is that it is a 'medium complexity' model that can be leveraged to study aerosol systems as a box model but too computationally expensive to be added to a chemical transport or chemistry climate model. I am inferring this from the model description. I would strongly encourage the authors to consider adding text that places UNIPAR as an appropriate model for the science to be addressed and how findings from UNIPAR can be used to build reduced-form models for atmospheric models.

*Response: Whereas UNIPAR has more complexity than SOA models typically used for atmospheric models, the computational cost of integrating UNIPAR into regional scale models is not prohibitively expensive. For example, UNIPAR was recently integrated into the CAMx model and used for air quality simulation (Yu et al., 2022). We have added discussion within the manuscript addressing this point:*

*(Section 3, Sentence 2)"Furthermore, UNIPAR has been recently integrated into the CAMx regional scale model and used to demonstrate SOA formation from various HCs including isoprene, terpenes, aromatics, and linear alkanes (Jo et al., 2023; Yu et al., 2022)."*

2) Table 1: If one is to peruse the molecules found in gasoline and diesel exhaust (one could also do this for oil and gas and volatile chemical products – other important sources for atmospheric alkanes), one would be hard pressed to find the types of molecules shown in Table 1. See, for instance, the work from Allen Robinson, Albert Presto, Drew Gentner, and Allen Goldstein between 2010 and 2020; I apologize for not listing other important contributors to this since I am going off of memory. Most branched alkanes seem to be much more 'branched' with a much smaller linear carbon backbone. A rationale is needed for why the species in Table 1 are relevant for the real atmosphere.

*Response: Ideally, a larger variety of branched alkanes should be used for a study like this but we are limited to branched alkanes which are commercially available in a pure form. However, the species we included spanned a relatively wide range of carbon numbers and branching structures, with 1-branched C12 to 7-branched C17 and 4-branched C19. Notably, highly branched Heptamethylnonane produced very little SOA mass. Even in the case that the atmosphere has significantly higher concentrations of branched alkanes with more branching and smaller linear carbon backbones than any of the compounds studied, these atmospheric compounds would likely be less relevant to atmospheric SOA concentrations due to negligible SOA yields compared to the compounds included in this study.*

3) Line 188: The comment that there are no explicit mechanisms for branched alkanes is not true. I see several branched alkane species in MCM (albeit not as heavily branched as those found in vehicle exhaust); see: https://mcm.york.ac.uk/MCM/browse. Similarly, I see pre-determined mechanisms for several branched alkanes for GECKO-A, suggesting that mechanism generation is possible for the branched alkanes studied in this work; see: https://www.acom.ucar.edu/gecko/output-library.shtml.

*Response: We agree that this sentence is too reductive and now has been updated for clarity. It now reads:*

*"For branched alkanes, explicit gas mechanisms are not currently available for the practical application to the UNIPAR SOA model, as described in the Introduction section."*

*In regard to the MCM, the branched alkanes with gas oxidation mechanisms available are too small (largest is C7 with 1 branch) to be relevant to SOA formation. For example, we find that C12 with 1 branch is barely able to form SOA. In regard to the GECKO-A mechanism, we describe the limitations in the introduction section that is now referenced in the corrected sentence above. GECKO-A is a model that can procedurally generate gas chemical mechanisms for a given structure. However, the number of oxidation products increases exponentially as the number of carbons in the precursor HC increases. Studies which used GECKO-A to create gas oxidation mechanisms for branched alkanes required severe simplifications to the mechanism in order for it to be feasible (Aumont et al., 2013; La et al., 2016). In regard to the link, it seems that this work can output the top 10 gas phase products for the set of precursors they included. However, that is not sufficient for populating our product array of 48 groups for a given precursor.*

4) 'Lumping array': My understanding is that MCM mechanisms were used to simplify the product distribution from oxidation of linear alkanes and adjusted for branched alkanes to create one of the key inputs (i.e., alpha?) to the UNIPAR model. This approach is very poorly described, and I encourage the authors to improve this. Also, how the mechanism is used to inform a dynamically varying product distribution using the aging factor is unclear.

*Response: To improve clarity, the following text has been added:*

*(Section 3.3, Paragraph 1)"During the process, each non-radical gas oxidation product of a specific precursor is lumped into one of the 48 species. During SOA simulations, all oxidation products within a specific lumped group with undergo partitioning, or particle phase reactions, as single species. By doing so, the UNIPAR model is able to leverage the complexity of a relatively large semi-explicit gas oxidation mechanism (generally between 200 to 500 non-radical species per precursor from MCMv3.3.1) while limiting the computational load."*

5) Line 230 onwards: It's unclear what the basis for the ARF rules is. Do these come from more fundamental studies that explore the probability of isomerization modulated by branching?

*Response: The basis for the ARF are now elaborated in Section S1.*

6) Equation 6: While I appreciate the treatment of separate phases based on the organic aerosol composition and varying RH, I was somewhat surprised to see Raoult's law being used directly to simulate the partitioning of organic species into an inorganic phase. Is this approach informed using more chemically-resolved models (e.g., AIOMFAC)?

*Response: The activity coefficient term is Eq. (6) is determined by Eq. (8). Eq. (8) is a semi-empirical equation that was derived from the AIOMFAC model as a function of molecular weight, O:C ratio, hydrogen bonding, sulfate fraction, and RH. Each lumping group will have a*

*unique molecular weight, O:C ratio, and hydrogen bonding term based on the products it contains. Sulfate fraction and RH are environmental inputs in the model.*

7) Figures 2-3: The model-measurement comparison is less than stellar and there are experiments where the model deviates significantly from the observations (12A, 16B). Since the results in Section 4.3 and beyond are based entirely on model simulations, how does one interpret the sensitivity results given the imperfect model comparisons in Figures 2 and 3. Are any of the model inputs/processes tweaked for the branched alkane results presented in this manuscript, relative to the linear alkane system? Can these tweaks, if they improve model performance, tell us something about mechanistic differences between linear and branched alkanes?

*Response: Regarding experiment 12A, the gas simulation showed a significant overprediction of HC consumption, likely due to the low purity (80%) of the Isododecane solution we used, so a poor agreement with model simulation is expected. Regarding experiment 16B, this gas simulation also shows an overprediction of HC consumption which leads to an expected overprediction of SOA mass production. The simulation also shows an underprediction of ozone formation, indicating that oxidation is underpredicted, which is contrary indication to the overprediction of HC consumption. This likely arises from the CB6r3 mechanism we use to predict HC consumption values as described in Section S1. Within the CB6 mechanism, alkanes are input piecewise as PAR, with 1 PAR representing 1 carbon. The reaction rate constant of PAR with OH is based on a set of concentrations of alkanes measured in the ambient atmosphere (Gery et al., 1988). Measured alkanes tended to be much smaller than the alkanes used in this study; smaller alkanes tend to have a larger proportion of primary carbons (OH reaction rate = 1.30E-13 $cm^3$/(molecule\*sec)) compared to secondary (OH reaction rate = 7.69E-13 $cm^3$/(molecule\*sec)) and tertiary (OH reaction rate = 14.90E-13 $cm^3$/(molecule\*sec)) carbons (Jenkin et al., 2018). Thus, the reaction of OH with PAR in the CB6 model is slower than the reactions happening during chamber experiments with larger alkanes. Within the CB6 mechanism, we include the following reaction for each HC in order to create an output for HC consumption:*

$$HC + OH = OH$$

*Essentially, this equation will predict HC consumption according to the CB6 simulated OH concentration at a given time. The fact that OH is not consumed within the above equation ensures that this has no impact on the underlying CB6 mechanism. If the PAR reaction with OH is too slow, then OH concentrations simulated by CB6 will be too high, which will lead to an overprediction of HC consumption according to the equation above. This would also explain the ozone deficiency within the gas simulation, suggesting oxidation is too slow within CB6 compared to this chamber experiment. Discussion regarding this issue has been added to section S1 and to the manuscript:*

*(Section 4.7, last paragraph)"Uncertainties also exist in hydrocarbon consumption values simulated by CB6. As described in section S1, the CB6 model overpredicts hydroxyl radical concentrations for chamber experiments performed with long-chain alkanes. Accordingly, gas simulations had a tendency toward overprediction of hydrocarbon consumption."*

*As far as model inputs and process, no specific changes were made to these components for branched alkanes. However, it is important to note, as we do in the conclusion, that the vapor*

*pressure of each branched alkane is not exactly the same as the linear alkane with the nearest vapor pressure. Thus, we do not expect the model to have perfect agreement with chamber data for any individual branched alkane. However, the model performance is reasonable enough that general trends for branched alkanes can be extracted with a higher degree of certainty. Furthermore, the parameters produced by this study are ultimately intended for application to regional SOA models. In the regional scale, HCs are typically input as classes of compounds rather than individual HCs being simulated. For example, Jo et al. (2023, ACP Preprint) simulates linear alkane SOA by splitting alkanes into 3 classes according to carbon number ranges. In future application for branched alkanes, a similar method will be used. Our parameters would be appropriate for use in this case as there is a higher level of acceptable deviation, in terms of vapor pressure, for a class of compounds than for any single compound. For example, there is a higher degree of uncertainty about the average vapor pressure of atmospheric set of branched alkanes ranging from C9-C13 compared to the degree of uncertainty of the vapor pressure of a specific branched alkane with a known structure.*

8) Sections 4.3-4.4: Do the results shown in Figure 5, qualitatively, align with those from chamber studies of branched alkanes (Lim and Ziemann, ES&T, 2009; Tkacik et al., ES&T, 2012; Loza et al., ACP, 2014)? Loza et al. (ACP, 2014) found that SOA mass yields were higher for alkanes under high NOx conditions, which contrasts with that shown in Figure 6. How does one reconcile these differences? How was the enthalpy of vaporization modeled for the SOA species, which should strongly influence the temperature sensitivity results shown in Figure 8? How important are oligomers in the end-of-experiment SOA? Is the low sensitivity to oligomerization rates shown in Figure 9 simply because oligomers don't account much for the total SOA? Overall, there are details missing in these sections that help place the results in context.

*Response :*

*Figure 5 does qualitatively agree with previous literature which also finds that an increase in branching leads to lower SOA yields. The following sentence has been added to the text to discuss this comparison:*

*(Section 4.3, Paragraph 1)"This result qualitatively agrees with previous literature which explore the impact of alkyl branches on SOA yields of alkanes (Loza et al., 2014; Lim and Ziemann, 2009; Tkacik et al., 2012)."*

*Loza et al. claims that alkane yields are higher, or the same, at high NOx conditions compared to low NOx conditions. However, after careful examination of their reported experimental results (seen below), their results do not clearly support this conclusion. HC yields can vary significantly according to the amount of precursor HC that is consumed. As demonstrated by Odum et al. (1996), higher $M_0$ values will lead to higher HC yields. As HC oxidation products partition to create SOA mass, this partitioned mass will act as $M_0$ for the reaction products of the next cycle, meaning that the oxidation products of each subsequent cycle will be more able to partition to form SOA mass. Thus, when comparing SOA yields for high and low NOx conditions, the comparison should be done with a similar precursor HC consumption. Generally, Loza et al.'s reported precursor HC consumptions were larger for their high NOx experiments than their low NOx experiments for the linear and branched alkane they studied (Methyl-undecane[Mud],*

*and Dodecane [Dod]) as seen in their experimental tables below. Furthermore, their chamber had consistently higher seed volumes for their High NOx experiments compared to their Low NOx experiments. For example, the highest seed volume for their low NOx experiments (21.8) was lower than the lowest seed volume in their high NOx experiments (26.1). This discrepancy further obscures the comparison of SOA yields between NOx conditions. Based on their experimental table, the most appropriate comparisons would be between MH1(ΔHC = 11.6 ppbv) and ML1(ΔHC = 8.4 ppbv), and DH1(ΔHC = 9.2 ppbv) and DL1(ΔHC = 7.9 ppbv). ML1, the low NOx experiment, has a higher yield than MH1. DH1 has a higher yield than DL1, but the yield for DL1 seems like an outlier within the context of this dataset. For example, DL1 has a significantly lower yield than ML1 with a similar amount of hydrocarbon consumption. It is highly unlikely that Methyl-undecane produces a significantly higher yield than Dodecane at a similar HC consumption and NOx conditions. This experimental result is contradictory to the broader literature which expects branched alkanes to have lower yields than linear alkanes with the same carbon number. Within our analysis of the impact of NOx on alkane yields, we perform sensitivity testing at a specific amount hydrocarbon consumption, seed condition (no seed), $M_0$, humidity, sunlight profile, and temperature for all simulations. Thus, we can isolate the impact of NOx in a simulation where other relevant factors are equivalent.*

*The following sentence has been added to the text to address Loza et al.'s study:*

*(Section 4.4, End of Paragraph 1)"This result is in agreement with Loza et al. (2014) who find a higher yield for Isododecane under low NOx conditions when similar amounts of hydrocarbon are consumed in both high and low NOx conditions."*

**Table 2.** High-NO$_x$ experimental details.

| Expt.[a] | Alkane | Seed vol. ($\mu m^3\,cm^{-3}$) | $NO_o$[b] (ppbv) | $NO_{2,o}$[b] (ppbv) | $HC_o$ (ppbv) | ΔHC (ppbv) | $\Delta M_o^c$ ($\mu g\,m^{-3}$) | Yield[c] (frac.) |
|---|---|---|---|---|---|---|---|---|
| MH1 | Mud | 31.7 ± 9.5 | 94.1 ± 0.5 | 6.6 ± 0.2 | 11.6 ± 0.4 | 11.6 | 8.5–17 | 0.11–0.21 |
| MH2 | Mud | 41.6 ± 12.5 | 97.7 ± 0.5 | 5.8 ± 0.2 | 79.6 ± 2.5 | 79.1 | 100–200 | 0.19–0.38 |
| DH1 | Dod | 30.9 ± 9.3 | 93.8 ± 0.5 | 6.3 ± 0.2 | 9.7 ± 0.3 | 9.2 | 19–40 | 0.30–0.62 |
| DH2 | Dod | 26.1 ± 7.8 | 96.8 ± 0.5 | 7.1 ± 0.2 | 59.2 ± 1.9 | 56.8 | 91–210 | 0.23–0.54 |
| DH3 | Dod | 30.4 ± 9.1 | 96.5 ± 0.5 | 6.1 ± 0.2 | 63.6 ± 2.0 | 61.2 | 100–210 | 0.22–0.51 |
| HH1 | Hch | 34.1 ± 10.2 | 101 ± 0.5 | 2.6 ± 0.2 | 11.5 ± 0.4 | 11.5 | 27–45 | 0.34–0.57 |
| HH2 | Hch | 40.0 ± 12.0 | 95.4 ± 0.5 | 2.9 ± 0.2 | 65.0 ± 2.1 | 64.9 | 210–270 | 0.46–0.61 |
| CH1 | Cdd | 38.7 ± 11.6 | 95.6 ± 0.5 | 6.8 ± 0.2 | 8.5 ± 0.3 | 8.5 | 56–91 | 0.98–1.6 |
| CH2 | Cdd | 37.7 ± 11.3 | 93.4 ± 0.5 | 7.9 ± 0.2 | 61.0 ± 2.0 | 58.6 | 320–400 | 0.80–1.0 |

**Table 3.** Low-NO$_x$ experimental details.

| Expt.[a] | Alkane | Seed vol. ($\mu m^3\,cm^{-3}$) | $HC_o$ (ppbv) | ΔHC (ppbv) | $\Delta M_o^b$ ($\mu g\,m^{-3}$) | Yield[b] (frac.) |
|---|---|---|---|---|---|---|
| ML1 | Mud | 21.8 ± 6.5 | 8.5 ± 0.3 | 8.4 | 7.9–15 | 0.14–0.27 |
| ML2[c] | Mud | 16.7 ± 5.0 | 28.9 ± 0.9 | 28.1 | 28–58 | 0.15–0.31 |
| ML3 | Mud | 15.9 ± 4.8 | 40.2 ± 1.3 | 38.1 | 49–86 | 0.19–0.33 |
| DL1 | Dod | 16.7 ± 5.0 | 8.2 ± 0.3 | 7.9 | 1.8–4.2 | 0.033–0.078 |
| DL2[c] | Dod | 12.1 ± 3.6 | 34.0 ± 1.1 | 33.6 | 35–65 | 0.15–0.28 |
| HL1[c] | Hch | 11.2 ± 3.4 | 15.6 ± 0.5 | 15.5 | 33–70 | 0.30–0.65 |
| HL2 | Hch | 20.0 ± 6.0 | 41.3 ± 1.3 | 40.8 | 99–120 | 0.35–0.44 |
| CL1 | Cdd | 18.9 ± 5.7 | 3.5 ± 0.1 | 3.4 | 4.9–11 | 0.22–0.46 |
| CL2[c] | Cdd | 15.3 ± 4.6 | 10.4 ± 0.3 | 10.3 | 30–62 | 0.42–0.86 |
| CL3 | Cdd | 21.5 ± 6.5 | 46.6 ± 1.5 | 45.1 | 200–230 | 0.61–0.73 |

*Figure. Experimental tables from Loza (2014, ACP)*

*The enthalpy of vaporization for each volatility group is now reported in Section 3.3 as follows:*

*(Section 3.3, paragraph 1)"The αi array consists of 6 different reactivity levels (very fast (VF), fast (F), medium (M), slow (S), partitioning only (P), and multi-alcohol (MA)) and 8 different volatility levels (1E-08, 1E-06, 1E-05, 1E-04, 1E-03, 1E-02, 1E-01, and 1.0 mm Hg) based on*

*vapor pressure which represent 48 species. The 8 volatility levels have enthalpy of vaporization values 140E+3, 106E+3, 96E+3, 89E+3, 82E+3, 58E+3, 58E+3, and 58E+3 J/mol, respectively."*

*Particle phase reactions in the UNIPAR model are represented as oligomerization. Thus, when we say that particle phase reactions are a relatively small part of alkane SOA, it means that oligomers account for a relatively small part of alkane SOA. The following sentence discussing Figure 9 has been updated for clarity:*

*(Section 4.5, paragraph 1)"This reaffirms the previously discussed (Sections 4.2, 4.4) idea that particle-phase reactions (oligomerization reactions) serve a relatively small part in branched alkane SOA formation due to the lack of reactive oxidation products."*

9) Section 4.6: I don't think the case study with diesel fuel is atmospherically relevant since diesel fuel rarely evaporates to emit alkanes. What could be useful is to use a speciation of diesel (or gasoline) exhaust to look at the relative importance of linear and branched alkanes. This analysis also misses the point that a lot of diesel fuel and diesel exhaust is comprised of branched-cyclic species, yields for which are relatively less certain than those for linear and branched alkanes.

*Response: We agree that a speciation of hydrocarbons in diesel or gasoline exhaust would be a more appropriate for this kind of analysis than fuels. Diesel fuel has been chosen to demonstrate the SOA potential from linear alkanes and branched alkanes in different carbon lengths using UNIPAR. Unfortunately, we are not currently aware of any diesel exhaust speciation papers that successfully speciate the branched alkanes present to the extent that is done by Gentner's diesel fuel speciation (i.e. relative concentration at different carbon numbers). Generally, branched alkanes are reported as a whole as "unspeciated branched alkanes". We have added some discussion that clarifies this point:*

*(Section 4.6, paragraph 1)"It is important to note that the composition of branched and linear alkanes in fuels will vary from the composition of those in fuel exhausts. However, literature which reports on the composition of fuel exhaust typically reports significant proportions of branched alkanes as unspeciated branched alkanes (Lu et al., 2018; Tkacik et al., 2014). Thus, current diesel exhaust compositions cannot be used for this analysis."*

*In regard to branched-cyclic compounds, these compounds were deliberately not included in this paper as the presence of cyclic structures significantly changes the types of products produced through oxidation. Generally, linear and branched alkanes are unlikely to form aldehyde functional groups while ring open reactions for cyclic alkanes commonly produce them. Because of this issue, the linear alkane parameters cannot be also extended to cyclic alkanes. Thus, alkane compounds with cyclic structures will be investigated separately in future studies. A note has been included to address the cyclic alkanes present in fuels:*

*"In addition to linear and branched alkanes, cyclic and branched-cyclic alkanes also represent significant proportions of fuels. The presence of a cyclic structure in a precursor alkane significantly changes the oxidation products it will create. Cyclic alkanes tend to produce oxidation products similar to linear alkanes except with an additional aldehyde group, which is*

*created during ring-opening reactions. The method used in this study to create branched alkane $\alpha_i$ arrays in this study cannot be extended to cyclic alkanes as the presence of additional aldehyde groups will influence not only the volatility of the oxidation products but also the reactivity. To appropriately identify the relative SOA formation potentials of all alkanes present in diesel fuels, future studies should include cyclic and branched-cyclic alkanes in this analysis."*

Minor comments:

1) Line 29: The statement about biogenic VOCs being more important in the future needs backing. Also, are plant wax emissions of alkanes – assuming most of these are branched alkanes – are a significant source relative to other biogenic VOCs (isoprene, terpenes)?

*Response: After reviewing emissions trends, we find that anthropogenic NOx and SO2 emissions have a decreasing trend but anthropogenic VOC emissions have not decreased in the same manner. We have removed the claim that biogenic VOCs being more important for SOA formation in the abstract and manuscript.*

*We add discussion to address the relative importance of alkanes comparison to other biogenic VOCs emitted in biogenics:*

*"Alkanes do not represent a large proportion of biogenic emissions in comparison to other biogenic VOCs but they tend to be large, low-volatility compounds so they can still significantly contribute to SOA (Männistö et al., 2023; Lyu et al., 2017)."*

2) Introduction: The importance of branched alkanes as SOA precursors to the atmosphere could be better stated by presenting findings from the literature. The current discussion is too qualitative.

*Response: Some text has been added to the Introduction (Paragraph 1) to state the importance of branched alkanes quantitatively:*

*"For example, Song et al. (2019) reports from their review of various VOC sampling studies that alkanes represent between 40.3% to 74.4% of VOCs in a set collected from Houston, Mexico, and various urban cities in China."*

*"Within the set of alkanes in their study, Song et al. (2019) finds that branched alkanes represented 33% of alkanes in Langfang, China."*

*"This is consistent with Caplain et al.'s (2006) study which reported that linear and substituted alkanes represent significant proportions of both gasoline and diesel fuel exhaust at 6-18% and 18-31%, respectively."*

3) Line 104: Could density estimation methods (e.g., from H:C and O:C data) or previous literature be used instead to determine SOA density?

*Response: FTIR spectra could be used to extract H:C and O:C data. However, FTIR data was only collected for a few experiments. Many experiments did not produce sufficient SOA mass to collect FTIR spectra. Literature data specifically for branched alkanes is relatively sparse.*

*Additionally, literature studies which collect branched alkane SOA data vary by many factors: branched alkane structures, type of oxidant used, NOx concentration, seed type, seed concentration, humidity, temperature. Thus, extrapolation of previously collected density data to branched alkanes experiments found in this study is not feasible.*

4) Line 140: Please define 'functionality distributions'.

*Response: This term has been removed from this sentence. Typically, we can use UNIPAR to identify the types of functional groups present in SOA as each lumped group is associated with a set of explicit products. However, this is not feasible in this case for branched alkanes as we do not have explicit mechanisms.*

5) Line 145: It's a little unclear if the lumping is performed for the precursor or oxidation products or both. Clarify what 'lumping array' means. I also think 'lumped array' is a better use of the term.

*Response: To alleviate any confusion, the term "lumping array" has been replaced by "$\alpha_i$ array" throughout the paper, with "lumped group" used to describe an individual component of the array and "lumping" used to describe the process by which the array is created.*

6) Line 183: Are there experimental or theoretical (e.g., molecular dynamics) studies that provide evidence for autooxidation pathways for linear alkanes? Is autooxidation also desirable for branched alkanes?

*Response: While there are no studies that document autoxidation reactions for alkanes specifically, previous literature regarding autoxidation has identified specific structures which may undergo autoxidation. During our addition of autoxidation reactions to the mechanisms of linear alkanes, we identified specific structures within the MCM mechanism that were documented to undergo autoxidation reactions within the literature.*

7) Equations 1-3: A rationale needs to be provided for this simplification in treating aging reactions and an example of how the aging kernel works. Citing previous work is necessary but not sufficient.

*Response:*

*Please find the added discussion regarding our aging kernel:*

*(Section 3.3, Equation 1-3) "Generally, the amount of oxidation within a given system is correlated with the concentrations radicals within the system, with higher concentrations of radicals in more aged systems. Thus, concentrations of major radicals, normalized by initial hydrocarbon concentration, are used to represent the amount of aging."*

*"Essentially, $[HO_2]$ and $[RO_2]$ are used to scale the oxidation product distribution of a given HC precursor (i.e. $\alpha_i$ array) between a fresh and aged composition."*

8) Sections 3.6-3.7: As most of these formulations to simulate particle-phase reactions and chamber artifacts come from previous work from this group, text needs to be added to

summarize if these formulations have been evaluated against direct observations and what the remaining uncertainties are.

*Response: Further discussion has been included in the manuscript to provide details on the previous demonstrations of these methods as well as remaining uncertainties:*

*(Section 3.6, paragraph 1)"The inclusion of particle-phase reactions has been demonstrated to significantly improve predictions for aromatic hydrocarbons (Im et al., 2014; Zhou et al., 2019). Particle-phase reactions were also included in the study by Madhu et al. (2023) which demonstrated a negligible impact on linear alkane SOA."*

*(Section 3.6, Equation 11)"As explained by Zhou et al. (2019), a significant uncertainty remains in the calculation of $[H^+]$ specifically in low RH and ammonia rich environments due to a poor performance of the E-AIM under these conditions (Li and Jang, 2012)."*

*"Han and Jang (2022) demonstrate this method which accounts for viscosity in their application to SOA predictions from gasoline vapor composed of aromatic hydrocarbons and long-chain alkanes."*

*(Section 3.7, last paragraph)"This method for correcting the bias originating from gas-wall partitioning has been previously demonstrated for toluene, TMB, α-pinene (Hang and Jang, 2020), and linear alkanes (Madhu et al., 2023), as well as a composition of gasoline vapor (Han and Jang, 2022). As explained by Hang and Jang (2020), uncertainties in this method are associated with the calculation of physicochemical parameters of lumped groups."*

9) Line 367: Is there literature evidence for this statement? If not, can you describe the mechanism?

*Response: Yes. Jang et al. (Science, 2002) describes the mechanism for the particle phase reactions of carbonyls. A citation has been added.*

---

## Author Response (AR2)

**Response to Editor**

Thank you for reviewing this paper. We hope we have appropriately addressed all of your concerns below.

1) The method of determining the functional group compositions of various alkane SOA using FTIR data were not described adequately in the manuscript. For example, what reference compounds were used? What peaks were used to quantify the different functional groups? How were the functional group compositions converted into O:C values? This type of information is needed to determine if the analysis is appropriate and for others to be able to reproduce the current measurements.

*Response: We have FTIR database from numerous chemical species which vary in functional groups and carbon lengths. FTIR data were collected from NIST database and some were measured from SOA related compounds (i.e., pinonaldehyde and pinoic acid).*

**Table S4.** *Peak assignments of FTIR spectra*

| Functional group | Peak ranges cm$^{-1}$ | O content | C content |
|---|---|---|---|
| OH stretching in alcohol | 3650-3200 | 1 | 0 |
| OH stretching in carboxylic acid | 3550-2500 | 2 | 1 |
| CH stretching | 2861, 2927, 2972 | 0 | 1 |
| C=O stretching in ketone and carboxylic acid | 1725 | 1 | 1 |
| C-O stretching (non-carboxylic acid and non-alcohols) | 1080-1240 | 1 | 1 |

*The area of each functional group is converted to the relative intensity of each functional group. The intensities are normalized by the intensity of CH stretching and used, alongside the values for Oxygen content and Carbon content, to calculate O:C values. This information has been added to Section S8.*

2) Response to Reviewer 2, Question 2. The response is good, but the authors should include some or all this text in the revised manuscript to address the Reviewers comment.

*Response: "Additionally, compounds used in this study, which were picked from a limited set of commercially available branched alkanes, are generally larger and less branched compared to those identified in the atmosphere and in fuel exhausts. However, even in the case that the atmosphere contains significant amounts of small, highly branched compounds, those would be less relevant for SOA formation potential than compounds used in this study due to negligible SOA yields."*

3) Response to Reviewer 2, Question 4. The authors have improved clarity partially, but the description of the lumping array method still remains unclear. Please expand the discussion and consider using examples to improve the clarity.

*Response: Discussion has been added to improve the clarity of the lumping array method description.*
*(Section 3.3)"For example, consider the compound 3-dodecanol, a product of n-dodecane photooxidation. 3-dodecanol has a calculated vapor pressure of 5.1E-3 mmHg (Myrdal and*

*Yalkowsky, 1997; Zhao et al., 1999; Stein and Brown, 1994). Because the vapor pressure is smaller than that of group 7 (1E-2 mmHg) and larger than that of group 6 (1E-3 mmHg), 3-dodecanol is lumped into volatility group 6. Due to a lack of functional groups which promote particle phase reactions (i.e. ketone and aldehyde), 3-dodecanol is lumped into reactivity group P and will only participate in partitioning. This process is performed to lump each non-radical gas oxidation product, with a sufficiently low vapor pressure, into one of the 48 lumped groups. If group 6P contains only 3-dodecanol, then the α value of that group represents the units (i.e. $\mu g/m^3$) of 3-dodecanol produced per unit of precursor HC consumed. If group 6P has multiple compounds, then then the α value of that group represents that the sum of the units produced of all products classified under 6P per unit of precursor consumed. After the initial lumping process, the amounts of each gas oxidation species produced per unit of HC consumed is extracted via a standardized gas simulation. This gas simulation predicts gas oxidation product concentrations for a given initial precursor concentration at various $NO_x$ levels under a sunlight file near the middle of summer (06/23/18). These simulation results produce lumped arrays for 4 different conditions: high $NO_x$ fresh, high $NO_x$ aged, low $NO_x$ fresh, and high $NO_x$ aged. The α value of each group is represented as a polynomial equation which is a function of the HC ppbC/ $NO_x$ ppb level (Table S3). After α values are calculated for a specific HC ppbC/ $NO_x$ ppb level, they can be scaled between fresh and aged compositions as described below."*

**Response to Reviewer 1**

Thank you for the additional comments. We hope we have appropriately addressed all of your concerns below.

1) Figures S2 Ozone: While the authors have corrected the ozone data for C19B, they have not addressed the concerns regarding C16B. Given that C16B exhibits higher NOx and HC levels compared to C16C, it is expected that C16B would show higher ozone levels than C16C according to the ozone isopleth. This is consistent with chamber data. However, the simulation significantly underestimates ozone for C16B, leading to much lower ozone levels than those of C16C. This discrepancy raises questions about the possibility of using incorrect data again or potential errors in the simulation.

*Response: After closer review, we found that the sunlight intensity data was delayed by an hour due to daylight savings time. Additionally, initial HC concentration in the simulation was increased slightly to better represent the GC-FID measurements. The prediction of Ozone has been improved significantly but overconsumption of HC in the simulation has been increased.*

2) L531: I recommend removing this sentence since the discussion on temperature dependency, in relation to Figure 8, has been removed.

*Response: The discussion on temperature dependency has been removed in regard to its impact on the contribution of autoxidation products to SOA. However, Figure 8 still shows a significant temperature dependency for branched alkane SOA as a whole and this discussion is included within the text:*

*(Preceding Fig. 8) "As expected, due to the relatively high importance of partitioning, the SOA yields of all the 3-branched alkanes are significantly impacted by changes in temperature under both $NO_x$ conditions."*

3) I recommend that the authors incorporate the comparison between the studies of Kwok and Atkinson (1995) and Jenkin et al. (2018), already detailed in their response, into the paper (supplement). Including this analysis would provide readers with an updated perspective on the recent developments on this subject.

*Response: Rate constants calculated using the Jenkin et al. (2018) framework have been added to table S1 under section S3.*

*Some discussion has also been added in section S3:*
*"Rate constants calculated using the framework by Kwok and Atkinson (1995) were used for gas simulations in this study. For comparison, rate constants calculated by a more recent framework (Jenkin et al., 2018) are included and do not show significant differences."*

4) What is the source of the enthalpy of vaporization values used in this study? The authors need to cite references or explain the basis for these values. While I agree that these values should increase as volatility decreases, it's not clear why the last three bins have the same enthalpy of vaporization values.

*Response: The enthalpy of vaporization and vapor pressure of homologous series of alkanes were used to produce a regression equation to predict the enthalpy of vaporization at given vapor pressures in the SOA model. The three highest volatility bins have the same values because, after a certain vapor pressure, further increases do not result in a significant change in enthalpy of vaporization values. This is demonstrated in the figure below. Enthalpy of vaporization values were taken from the study by Chickos and Hanshaw (2004). A description has been included in section S8.*

[Figure]

**References**

Myrdal, P. B. and Yalkowsky, S. H.: Estimating Pure Component Vapor Pressures of Complex Organic Molecules, Industrial & Engineering Chemistry Research, 36, 2494-2499, 10.1021/ie950242l, 1997.

Stein, S. E. and Brown, R. L.: Estimation of normal boiling points from group contributions, Journal of Chemical Information and Computer Sciences, 34, 581-587, 1994.

Zhao, L., Li, P., and Yalkowsky, S. H.: Predicting the Entropy of Boiling for Organic Compounds, Journal of Chemical Information and Computer Sciences, 39, 1112-1116, 10.1021/ci990054w, 1999.